# Predicting Network Motif Fingerprints with Graph Neural Networks

## Abstract

Graph Neural Networks (GNNs) are a predominant method for graph representation learning. However, beyond subgraph frequency estimation, their application to network motif prediction remains underexplored, with no established benchmarks in the literature. We propose to address this problem, framing motif estimation as an extension of subgraph frequency estimation. Our approach formulates motif estimation as a multitarget regression problem, optimising for interpretability and improving stability and scalability on large graphs. We validate our method using a large synthetic dataset generated by graph generators that mimic real-world data, and further test it on real-world graphs. Our experiments reveal that 1-WL limited models trained on synthetic data struggle to predict accurately motif profiles of real-world networks. However, apart from their reasonable performance within synthetic data, they can generalise to approximate the graph generation processes of real-world networks by comparing their predicted motif profiles with the ones originating from synthetic data. This first study on GNN-based motif estimation sets a benchmark and should open pathways for further developing the connection between motif profiles and subgraph frequency from a graph representation learning perspective.

## 1 Introduction

A structure is called a network motif when its recurring occurrence is not solely explained by randomness. These structures are extremely powerful tools for understanding complex networks. Understanding what substructures are relevant and not relevant to a graph can help understand the fundamental organisational principles behind it. This understanding enhances theoretical knowledge of network structure and function but also has practical implications in various fields, particularly in Biology. For instance, the feed-forward loop has been identified as a crucial functional pattern in many real biological networks of gene regulation (Mangan & Alon, 2003). It has also been discovered that motifs enable efficient communication and fault-tolerance across transcriptional networks (Roy et al., 2020). Furthermore, the related concept of graphlet degree distribution – a generalisation of degree distribution to higher-order structures – has been used to understand what is a good network model for protein-protein interactions (Pržulj, 2007).

Discovering a motif entails counting the number of occurrences of the desired structure, both in the network in study and in a set of control networks to understand its significance. However, this process is a very hard computational task. Just determining if a subgraph exists in a larger network (subgraph isomorphism) is a NP-complete problem (Cook, 1971). Even though methods to perform an analysis based on motifs exist (Ribeiro et al., 2021), they have high temporal complexity, rendering them intractable for very large networks. Furthermore, methods that rely on machine learning to address this problem typically do not give very interpretable results, a critical concept when doing analysis based on the relative importance of substructures.

**Present Work.** We aim to design a method for motif finding, leveraging a novel formulation that hinges primarily on reworking the target task to something else other than direct substructure counting. Our approach focuses on providing highly interpretable scores, ensuring the possibility of further insight into the conclusions obtained. Additionally, our method is robust and versatile, capable of operating effectively on graphs of any size. Knowing how difficult it is to obtain a high volume of real-world graph datasets that have both high quality and variety, we create a large synthetic dataset,

employing a myriad of generators and using it as the training data to assess the efficiency of our formulation. This setup leads to the conceptualisation of the following research question. Can Message Passing Neural Networks (MPNNs) under a different problem formulation than graph counting, be enough to accurately predict motifs of real-world graphs when trained on synthetic data?

**Key Contributions:** Our key contributions can be summarised as follows: *(1)* We show that the difficulty of motif discovery with MPNNs can be manipulated through different formulations of the target variable. Different formulations pertain to different concepts of motifs. Hence, depending on the the concept used, motif estimation does not have to follow the limitations from the literature regarding subgraph counting with MPNNs; *(2)* We make available a large diverse synthetic dataset in terms of graph topology and motif significance-profile generated with 23 synthetic generators. We also present a collection of more than 100 real-world networks and their motif significance-profile; *(3)* Our experiments show that MPNNs trained under the adopted motif concept with synthetic data can predict the significance-profile of synthetic data with a solid accuracy. Furthermore, we show they cannot generalise well for real-world data, showing a gap between these types of networks.

## 2 PRELIMINARIES

Let a graph $\mathcal{G} = (\mathbb{V}_{\mathcal{G}}, \mathbb{L}_{\mathcal{G}}, \boldsymbol{X})$ where $\mathbb{V}_{\mathcal{G}}$ denotes the vertex set of $\mathcal{G}$, $\mathbb{L}_{\mathcal{G}} \subseteq \mathbb{V}_{\mathcal{G}} \times \mathbb{V}_{\mathcal{G}}$ the edge set of $\mathcal{G}$, and $\boldsymbol{X} \in \mathbb{R}^{d_1 \times d_2}$ the vertex features such that $\forall v \in \mathbb{V}_{\mathcal{G}}, \boldsymbol{x} \in \mathbb{R}^{d_2}$. Let all edges by undirected such $(u, v) \in \mathbb{L}_{\mathcal{G}} \Leftrightarrow (v, u) \in \mathbb{L}_{\mathcal{G}}$. Let $\mathcal{H}$ be a subgraph of $\mathcal{G}$ if and only if $\mathbb{H}_{\mathcal{H}} \subseteq \mathbb{V}_{\mathcal{G}} \wedge \mathbb{L}_{\mathcal{H}} \subseteq \mathbb{L}_{\mathcal{G}}$, such that exists an injective homomorphism given by the injective function $f : \mathbb{V}_{\mathcal{H}} \mapsto \mathbb{V}_{\mathcal{G}}$ such that $(v, u) \in \mathbb{L}_{\mathcal{H}} \Rightarrow (f(v), f(u)) \in \mathbb{L}_{\mathcal{G}}$. If $f$ is bijective and $f^{-1}$ is an homomorphism (injective by construction) the relation is an isomorphism and the subgraph induced.

## 3 RELATED WORK

In order to discover motifs, we must define three steps: (1) What is the set of graphs, $S_G$, that we admit as candidates for motifs; (2) What method is used to count the occurrences of graphs of $S_G$ in the graph of interest $\mathcal{G}$; (3) How is the significance of the obtained counts calculated.

### 3.1 STEP ONE - HOW TO DEFINE THE SET OF GRAPHS USED

In this step, $S_G$ is typically defined *a priori*. This method is the most widely used (Milo et al., 2004a;b; Shen-Orr et al., 2002), and in most common cases, the selection of graphs used are ones known to be important to the area of the work in question (Shen-Orr et al., 2002; Alon, 2007).

Defining $S_G$ *a priori* is frequent for "non-machine-learning" techniques, but it is also common in machine-learning ones (Rong et al., 2020; Ying et al., 2020). However, when using techniques based on machine-learning, it is easier to create a task that can infer structures in $\mathcal{G}$ than when using non-machine learning approaches. Hence, motif discovery can be modulated as the task of finding the best set of graphs that are motifs according to a defined graph metric. That is, discover what graphs exhibit a certain criteria in order to be considered motifs (Bénichou et al., 2023; Zhang et al., 2020).

### 3.2 STEP TWO - HOW TO COUNT SUBGRAPHS

**Non-GNN Methods.** Numerous methods exist for approximating subgraph counts, eschewing dependence on Graph Neural Networks (GNNs) or any machine learning techniques. We refer the interested reader to Ribeiro et al. (2021) for a survey of these methods.

**GNN Methods.** Counting occurrences of a graph $\mathcal{G}$ in another graph $\mathcal{H}$ using GNNs was first introduced by Chen et al. (2020). This work introduced significant limitations of what substructures MPNNs can count. Subsequent works have refined MPNN-like models to be more expressive, allowing them to have guarantees of being able to count occurrences of more graphs. One branch of such models is known as node-rooted Subgraph GNNs. These will extract a $k$-hop neighbourhood for each node in the graph to be studied. Since they act per node and add a feature to the node that induced each subgraph, they are called node-rooted Subgraph GNNs.

These architectures, with models as powerful as 1-WL as the backbone, are strictly more powerful than maximum powerful MPNNs but are less powerful than the 3-WL test (Frasca et al., 2022; Yan et al., 2023; Zhang et al., 2023). Hence, they have limitations regarding the type of structures that they can count. Huang et al. (2022) gives a characterisation of what substructures Subgraph GNNs cannot count at node-level based on the notation of cycles and paths. They show that the Subgraph GNNs cannot count cycles of four or more nodes and paths of three or more nodes.

On a similar note, Huang et al. (2022) propose extracting edge-rooted subgraphs rather than node-rooted and marking nodes that form the edge that anchors the subgraph extraction. Huang et al. (2022) prove that the utilisation of double marking grants enhanced computational capabilities compared to node rooted Subgraph GNNs. Additionally, it is ascertained that their model exhibit partial superiority over the 3-WL test, enabling the counting of cycles with lengths shorter than seven and all subgraphs up to size four in a non-induced setting.

Recently a new theoretical view of Subgraph GNNs based on the Subgraph Weisfeiler-Lehman, a new version of the WL test, has been proposed (Zhang et al., 2023). This analysis presents a characterisation of the expressive power of all node-rooted Subgraph GNNs. They conclude that no node-rooted Subgraph GNN can be more powerful than the 2-folklore-WL (3-WL). This bound was already discussed by Yan et al. (2023) and Frasca et al. (2022). However, Zhang et al. (2023) demonstrate that no node-rooted Subgraph GNN can achieve the maximum expressivity of their time complexity class. This result draws a limitation in the design of node-rooted Subgraph GNNs. Later work by Yan et al. (2023) characterise the counting power of Subgraph GNNs for general architectures and a general number of rooted nodes used as backbone. Furthermore, a method to compare the expressivity of GNN models was introduced by Zhang et al. (2024) based on homomorphisms, they summarise the ability of multiple GNN models on their ability to count any substructures with no more than eight edges and no more than six vertices. Regarding induced subgraph counting at graph level, the subject of our work, 1-WL models cannot count any pattern with 3 or more nodes.

### 3.3 STEP THREE - HOW IS SIGNIFICANCE OBTAINED

After obtaining the frequency of the structures in $S_G$, the next step is to evaluate their significance. Hence, it is necessary to have an idea of what would produce, with no factor other than random chance, a network similar to $\mathcal{G}$ for some characteristic of interest. Let us denote NULL as a model that can achieve that goal. One example of NULL is a model that, given a graph $\mathcal{G}$, randomly switches edges while keeping the degree distribution of $\mathcal{G}$ – degree distribution is the characteristic of interest. The rewiring process is completely random and without any bias towards any predisposition (Milo et al., 2004a;b).

### 3.4 MOTIFS AND GRAPH NEURAL NETWORKS

Motif estimation, when approached through the lens of GNNs, appears to be a challenge that, to the best of our knowledge, remains largely unexplored in the existing literature.

**Directly counting.** One of the approaches that better matches direct motif estimation with GNNs would be to count subgraphs of interest, $S_G$, using a GNN in the input graph $\mathcal{G}$ and, after selecting a suitable null model, generate an amount $T$ of graphs according to it and use the same GNN model used in $\mathcal{G}$ to count the occurrences of the selected set of subgraphs in each of the $T$ control graphs.

**Motifs as tool.** Other works that integrate GNNs and motifs typically deviate from estimating motifs and use pre-computed ones to enhance the power of GNNs. Examples of this work include Motif Convolutional Networks (Lee et al., 2018), motif2vec (Dareddy et al., 2019), Motif Graph Attention Network (Sheng et al., 2024), Motif Graph Neural Network (Chen et al., 2023) and Heterogeneous Motif Graph Neural Network (Yu & Gao, 2022). In the field of GNNs, another usage of the concept of a motif as a relevant pattern comes from the attempt to explain the decision of GNN models. Two examples in this field are GNNExplainer (Ying et al., 2019) and TempME (Chen & Ying, 2023).

### 3.4.1 LEARNING RELEVANT PATTERNS

We will introduce the forthcoming studies under the term "relevant patterns" since most of them use a definition of motif that is different from the one we introduce. Nonetheless, when discussing such works, we follow the terminology of the original works and will call the pattern "motif".

Works directly pertaining to motif estimation (used to refine a downstream task) are MICRO-Graph (Zhang et al., 2020) and MotiFiesta (Oliver et al., 2022). An example of a method made to estimate subgraph frequency is SPMiner (Ying et al., 2020).

The main problems of these works are the following: (1) either the model does not assume a null model and returns raw counts of occurrences of a general $\mathcal{H}$ in $\mathcal{G}$ (SPMiner and other frequency estimation models), or (2) the model may use a null model to guide motif search but only returns the subgraph(s) that are considered a motif by the model, meaning it is typically not possible to query for a specific $\mathcal{H}$ (MICRO-Graph and MotiFiesta). Hence, these models typically ignore everything not branded as a motif, sometimes not even returning a motif score for the graphs regarded as such. Furthermore, models that return the raw count of occurrences can suffer from poor generalisation since the number of graph structures grows super-exponentially (Fu et al., 2023). As the size of a graph $\mathcal{G}$ grows, the possible counts of a substructure $\mathcal{H}$ in $\mathcal{G}$ also grow super-exponentially, causing high variation between results of small and very-large graphs. This fact can hinder the learning process of models that aim at being agnostic of network size and topology. Also, for raw count models, since no null model is assumed, obtaining significance implies added subsequent computation.

## 4 INITIAL PROBLEM DESCRIPTION

Hereafter, referencing the number of occurrences of a graph $\mathcal{H}$ within $\mathcal{G}$, denotes the induced count of $\mathcal{H}$ in $\mathcal{G}$. Furthermore, all graphs are undirected and they do not have edge features.

According to the definition of motif adopted, to understand if a graph $\mathcal{H}$ is a motif of a graph $\mathcal{G}$, we must know the number of occurrences of $\mathcal{H}$ in $\mathcal{G}$. Let us denote such count as $C(\mathcal{H}, \mathcal{G})$. Furthermore, to grasp the importance of $\mathcal{H}$ in $\mathcal{G}$, it is needed to know the count of $\mathcal{H}$ across sufficient graphs derived from a null model denoted as NULL. Let us denote the average count of $\mathcal{H}$ in graphs derived from NULL as $C^\mu(\mathcal{H}, \mathcal{G}_{\text{NULL}})$ and the standard deviation as $C^\sigma(\mathcal{H}, \mathcal{G}_{\text{NULL}})$. Hence, $Z(\mathcal{H}, \mathcal{G}_{\text{NULL}}) = \frac{C(\mathcal{H}, \mathcal{G}) - C^\mu(\mathcal{H}, \mathcal{G}_{\text{NULL}})}{C^\sigma(\mathcal{H}, \mathcal{G}_{\text{NULL}})}$ denotes the standardization (Z-score) of the occurrences of a graph $\mathcal{H}$ in $\mathcal{G}$.

### 4.1 OUR APPROACH

Even though not used for subgraph counting, we implant degree features into the graphs to enhance the capability of the models. Instead of modelling the learning task as predicting a single value $Z(\mathcal{H}, \mathcal{G})$ for some $\mathcal{H}$ and some $\mathcal{G}$, we model it as a multi-target regression problem in order to predict the motif score of multiple subgraphs at once. This characterisation naturally allows the construction of motif fingerprints as proposed by Milo et al. (2004b). Thus, we start with a vector of Z-scores, $\boldsymbol{z} = [Z(\mathcal{H}_1, \mathcal{G}) \ldots Z(\mathcal{H}_n, \mathcal{G})]$.

The restriction of the number of graphs in $\boldsymbol{z}$ implies that the proposed model will not be able to search if an arbitrary graph is or is not a motif. However, by having a model that has a more restricted objective, we aim to achieve higher precision in the said objective. Since $\boldsymbol{z}$ has a restricted size, one other aspect that deserves careful consideration is deciding what is the size of $\boldsymbol{z}$ and what graphs compose it. Should the selected graphs exhibit negligible relation, an attempt to predict the Z-score concurrently for all graphs may prove harmful to the performance of the model. In this case, such an approach forces the model to incorporate distinct patterns to predict scores for each graph, thereby resulting in a sub-optimal global predictive efficacy. However, if $\boldsymbol{z}$ is composed of a well-thought group of graphs, allowing them to share common patterns from a learning perspective, we hypothesise that the performance of the model can improve when compared to predicting just one Z-score, due to the possibility of what is learned about a target variable be "shared" with others through weight sharing (one other advantage is the need to only train a single model instead of multiple). A good candidate for $\boldsymbol{z}$ should have patterns that are interconnected with each other, either from the point of view of the Z-score distribution or from a topological one.

Building on top of what was described in the two previous paragraphs, we focus on small graphs, in particular all connected graphs of size three and four. This is also supported by existing relevant literature (Milo et al., 2004b;a; Shen-Orr et al., 2002; Asikainen et al., 2020; Pržulj, 2007; Ribeiro & Silva, 2013) suggesting that to understand a complex network, it is an important to understand how small graphs behave in that network. We focus on these graphs because their proximity in size should allow them to have a topological connection that translates in a connection in their Z-Scores.

Restricting the size of the graphs used in $z$ to small ones also has the added benefit that we can get the ground-truth of motif scores for a diverse type and size of networks, allowing for a richer train dataset. Furthermore, we expect that using a set of graphs of increasing size in the number of nodes and edges gives enough interconnectivity between their patterns from both a topological and a Z-score distribution point of view to allow the model to have a strong inductive bias towards meaningful patterns, allowing for a stronger performance. For example, a graph with many size four cliques will probably have a small amount of 4-stars.

We will refer to the set of graphs used to populate $z$ as $\Omega$. The function notation $\Omega(\mathcal{G})$ gives the set of all graphs that have the same number of connected nodes as the graph $\mathcal{G}$. For the chosen graphs it is possible to create two groups in $\Omega$, the graphs of size three and the graphs of size four. Using the motif Z-score directly in the learning task not only makes the function and result highly interpretable, but also eliminates the need to compute multiple networks based on the NULL model to determine significance. By normalising the Z-score across groups of graphs, as defined in Milo et al. (2004b), the values of $z$ are constrained between $-1$ and $1$, independent of network size, enhancing the model's predictive stability across various network scales. Moreover, normalising the Z-scores with $s_i = z_i/(\sum_{j \in \Omega(i)} z_j^2)^{1/2}$ imposes a mathematical interconnectivity between the Z-scores of graphs of the same group. This relationship, where the sum of squared Z-scores equals 1, supports a multi-target objective and further strengthens the problem formulation by adding an additional layer of interdependence among graphs. The normalised Z-score refers to the values of the significance-profile $s$. The learning task consists of minimising the MSE between the true and predicted significance-profiles.

## 4.2 ON THE RELATION WITH EXPRESSIVITY THROUGH SUBGRAPH COUNTING

It is expected that the expressivity regarding substructure counting to be highly related to the expressivity of discovering the significance-profile of graphs. Concretely, the problem of counting graphs is a subset of the problem of discovering significance-profiles where reducing the null model to nothing reduces the problem to graph counting.

Since $P$, the problem of counting graphs, is a subset of $S$, the problem of significance-profile estimation, it is possible to obtain instances of $S$ that are as hard as $P$, easier than $P$ and harder than $P$. Under the assumption that $S$ and $P$ function around the same set of graphs, these differences in difficulty come from the choice of null model. In the case of $S = P$, the null model should do nothing, for example, returning always $0$. In the case of $S < P$, the null model could always return the counts of each subgraph in a graph $\mathcal{G}$ without modifying $\mathcal{G}$, reducing the problem to always predicting a vector of zeros. For the case of $S > P$, employing a null model that randomly returns counts for $\mathcal{G}$ should make the problem theoretically harder since the model would have to learn the random process employed to correctly construct the significance-profiles. Thus, theoretical guarantees of expressivity might not hold depending on the selected null model. For instance, a recent demonstration solved the dimensionality of the $k$-WL test for induced subgraph count, stating that to perform induced subgraph count of any pattern with $k$ nodes we need at least dimensionality $k$ (Lanzinger & Barceló, 2023). Furthermore, no induced pattern with $k + 1$ nodes can be counted with dimensionality $k$, a result not verified to non-induced counts (Lanzinger & Barceló, 2023). However, depending on the null model, when working with significance-profiles over graphs of size four in $\Omega$, it might not be enough to have a model as powerful as the $4$-WL to guarantee enough power to correctly identify the normalised Z-scores of the size four graphs in $\Omega$.

**Modifications made to the problem.** In Section 4, we attempted to reduce the difficulty of the problem by formulating it as a multitarget regression of interconnected values from an algebraic and topological point of view. However, even in the case where it exists a perfect dependence between graphs of the same group of $\Omega$, or even across different groups, the problem is still at least as hard as finding the significance-profile of the graph(s) that governs the dependence relation. For example, if the significance-profile of the 3-path was symmetric to the significance-profile of the 3-clique, it would still be necessary to determine the significance-profile for 3-paths to compute the significance-profile of 3-cliques and vice versa.

**Testing with 1-WL bounded models?** MPNNs cannot perform induced counts of patterns of three or more nodes (Chen et al., 2020). Nevertheless, MPNNs are not inherently incapable of counting patterns in any graphs. Rather, for a pattern $\mathcal{H}$, there exists graphs $\mathcal{G}_1, \mathcal{G}_2 \in \mathcal{G}$ such that $C(\mathcal{H}, \mathcal{G}_1) \neq$

$C(\mathcal{H}, \mathcal{G}_2)$ and for any MPNN $M$ under 1-WL, $M(\mathcal{G}_1) = M(\mathcal{G}_2)$. Hence, $M$ cannot discover $C(\mathcal{H}, \mathcal{G}_1)$ and $C(\mathcal{H}, \mathcal{G}_2)$ simultaneously. However, within the 1-WL framework, MPNNs remain highly valuable and find numerous practical applications in real-world scenarios. Furthermore, we did not construct any characterisation of the problem space of $S$ regarding $P$ for the null model used. Hence, we might have made the problem easier (or harder) than substructure counting. The same applies to the usage of a multi-target objective. Thus, we will scrutinise the capability of MPNNs to address our particular challenge. Furthermore, more expressive models like Subgraph GNNs have a very high computational complexity, being very hard to use in large-scale graphs.

## 5 DATASETS

We rely exclusively on synthetic graphs, rather than commonly used GNN datasets such as Wu et al. (2018); Gómez-Bombarelli et al. (2018); Morris et al. (2020); Hu et al. (2021b;a). Our goal is to have a large dataset with diverse topological features, which would be difficult and costly to obtain from real-world data spanning multiple domains. Using multiple graph generators designed to simulate real-world phenomena and fully exploring their parameter space, we aim to create a dataset with both high topological diversity and a close resemblance to real-world data. Additionally, synthetic data allows for flexible expansion of the dataset size. Using synthetic data to train GNNs is not a new concept. However, as far as we know, most of the popularly used datasets typically have very small graphs (at most few hundreds of nodes) and are generated from a small set of generators, often random regular graphs and Erdős-Renyi graphs (Chen et al., 2020). Another popular type of synthetic graph dataset for benchmarking is small handcrafted graphs to limit test GNN models (Abboud et al., 2021; Murphy et al., 2019; Balcilar et al., 2021; Wang & Zhang, 2023). While still very limited, the only exceptions identified are Veličković et al. (2020); Corso et al. (2020).

Since we are not interested in limit testing the power of the model in comparison to theoretical tests, and instead aim at having a diverse dataset regarding graph topologies and motif scores, we create a new dataset using a total of 23 synthetic generators (11 non-deterministic, 12 deterministic). We explore their graph-generating space in order to extract all types of possible topologies they can express while limiting the graph size in order to avoid bottlenecks that increase training time beyond what we find reasonable for the amount of time and resources available. The final dataset puts the non-deterministic segment with 109164 graphs, and the deterministic segment with 38400 graphs, totalling for $\approx 250$ million nodes and $\approx 750$ million edges. Section A has a description with greater detail on how each generator was explored. For the ground-truth, we calculate $s$ using G-Tries (Ribeiro & Silva, 2013).

Exploring the myriad of significance-profiles from the generated graphs leads us to the conclusion that the 3-path and the triangle can only take on a few sets of values, being those $\{-0.707106, 0, 0.707106, 1\}$ for the 3-path and the first three values for the triangle. This leads to a strict result on the inter-variability of their Z-scores. Apart from cases where both scores are zero or the 3-path is 1 and the triangle is 0 (an artefact from the G-Trie model, primarily affecting the duplication-divergence model), we found that the Z-scores for graphs of size three are symmetric. This means that if one structure takes a Z-score of $x$, the other will take $-x$. In normalised form, these values are mapped to $0.707106$ and $-0.707106$, respectively. For the size four graphs, we can say that the size three encodes some information about the significance-profile of graphs of size four, alluding to the possibility of an advantage of using graphs of both sizes in the target variable (Figure 3 - Appendix A.4 has a detailed view of this result). As for strict dependence between the graphs of size four, apart from the mathematical based constraint, we could not confirm any other.

**Real-World Data.** Since we are interested in assessing the performance of the models with real-world data, we compiled a test set based on real networks of multiple categories. Besides varying the type of network, we vary in their relative size in terms of number of nodes and edges. We devise two categories based on the size of the networks used in the train set: (1) small-scale networks, (2) medium-large scale networks. Section A.5 presents a detailed description of the networks collected.

## 6 METHODOLOGY

The model used in the experiments is a very simple model similar to the one described in Chen et al. (2020) definition A.1. from Appendix A. Section B further details the model and how it was trained.

**Baselines.** We define a baseline denoted by an horizontal red line corresponding to the expected loss incurred when predicting the significance-profile $s$ by employing an independent random uniform model for each component of $s$. A second base line denoted by an horizontal blue line corresponds to the expected loss when using a model predicting a random value for every component of $s$, but taking into account the restrictions posited in Section 4 regarding the range of values each group of $\Omega$ can take. A final baseline dependent on the architecture of the model is represented by a light pink bar. It illustrates the mean error derived from using the model in question with random weights.

**Persistent Patterns.** Given a set of significance-profiles $\mathbb{S}$, we define the persistency $\rho$ of a pattern $s \in \mathbb{S}$ as the frequency of its occurrence within $\mathbb{S}$. The higher the number of occurrences of $s$, the more persistent the pattern will be. Considering the randomness inherent to deriving a significance-profile, we use a threshold, $t$, to decide what patterns are equivalent to each other. Given $t$, discovering the persistence of $s$ is reduced to $\rho_s = |\{s' \mid d_\infty(s, s') \le t, \forall s' \in \mathbb{S}\}|$ where $d_\infty$ is the Chebyshev distance. We employ the concept of persistent patterns to develop a high-level understanding of the nature of the true and predicted significance-profiles. We use the true significance-profiles as a standard to determine an appropriate threshold for equivalency. This threshold is then applied to the predicted significance-profiles. Section C further details the process to obtain $t$.

**"Correct" Predictions.** Since having a criteria for a prediction for a significance-profile to be correct/useful depends largely on the research field, in order to have a systematic threshold, we study how many predictions for each generator have all individual values of the predicted significance-profile below the obtained test error. Furthermore, we ensure that for a prediction to count as "correct", all significance-profiles for graphs in $\Omega$ have the correct sign. Presuming the test errors obtained for the synthetic dataset are relatively low, this approach guarantees that the predictions deemed "correct" not only contribute to a lower test loss than the one observed but are also potentially valuable from an application perspective. Let us denote a benchmark model as $T$. Let this model predict a random pattern according to the criterion used for the horizontal blue line described above. If we define a cutoff mean loss for $T$ of $0.25$ ($0.5$ absolute error), this model will incur in a possible error of $25\%$ of the maximum allowed. According to 1e8 simulations, $T$ will have a rate of "correct" guesses of $0.364\%$ while following the restrictions here introduced.

# 7 RESULTS

The chosen models correspond to one instance of GIN and SAGE for the non-deterministic segment and one instance of GIN for the deterministic one (see Appendix D for more details). Figure 1 shows the results for the selected models for the small and medium-large datasets. The yellow bar denotes the validation error, the blue bar represents the test error observed in the test dataset and the green bar the result of the model in the real-world dataset. The other marks represent the baselines from Section 6. A more detailed version of the predictions is available in Appendix D.2.

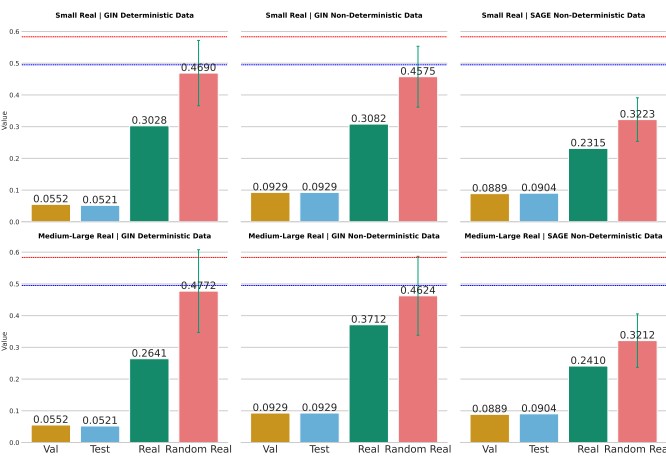

Figure 1: Scores of the best models for the test and real-world datasets.

## 7.1 Discussion of the Results

The models have low generalisation error as training, validation, and test errors are similar, and there is no significant difference in performance between GIN and SAGE for synthetic data. However, the models present a much higher error in the real-world dataset than the out-of-sample error estimated through the test set. Furthermore, the obtained value is dangerously close to the error obtained with random weights minus one standard deviation. Nonetheless, one positive is the stability of the error across the two different real-world dataset sizes. Even though the error is high, it remains stable as the size of the networks increases, a growth that exceeds three orders of magnitude in scale.

**What is the model not doing.** It is evident that any trained model does not adhere to an independent uniform random prediction and is not randomly predicting based on the restrictions of $s$. All predictions conspicuously surpass the red line and blue line, established as the initial benchmark.

**What can the model be doing.** Regarding the poor performance, one possible scenario is that the synthetic data used does not accurately reflect the real-world data, leading to the observed dissonance between scores in the test and real datasets. However, it is essential to consider the limitations of the MSE for this particular analysis. While MSE is valuable for comparing models with significant differences in performance, it does not provide sufficient information regarding individual predictions and their global shape. For instance, a model predicting a value of $0$ for all graphs would have an expected MSE of $0.25$. Using the concept of persistent patterns, we can understand how the predictions are distributed in terms of proximity to one another in relation to what would be expected from the ground-truth. Table 3 (Appendix C) conveys a detailed view of the clusters found for the threshold inferred from the true significance-profiles. Summarily, we conclude: (1) all models exhibit a substantially lower number of predicted patterns than expected; (2) this reduction is primarily attributed to the higher $\rho$ of select patterns, rather than an higher mean $\rho$ across patterns.

In the following sections we answer: (1) What can cause a model to tend towards persistent patterns? (2) Can we take the low error in the test set as a signal that the model is learning the synthetic data? (3) What causes the discrepancy between the scores in the synthetic and the real-world dataset?

## 7.2 Tendency for Persistent Patterns.

This study examines the impact of dropout on the expressivity of models and persistence of the patterns. By systematically increasing dropout values, the model's power is reduced due to regularisation. Testing with both GIN and SAGE models reveals that dropout on the MPNN has little to no effect on GIN's performance but slightly affects SAGE, indicating GIN's greater expressivity. This possibility was briefly discussed in Section 4.2, suggesting that less expressive GNN models are capable of distinguishing fewer graphs. The results suggest that models lacking expressivity generate fewer diverse patterns, leading to similar predictions across graphs. Hence, poor performance on real-world data might not solely result from limitations of the synthetic dataset, but also from the lack of expressivity of the models.

**Answering question (1).** Tendency towards persistency of patterns higher than what would be expected stems from the lack of expressivity of either the model, the dataset, or both.

## 7.3 Model Predictions in the Synthetic Dataset

Following Figure 8 (Appendix D.2), we conclude that the predictions made by all the models are reasonable for all generators. The results suggest that the model is sufficiently expressive to distinguish between different graph generators, as predictions often align with the true mean significance-profile. However, it struggles to differentiate graphs within the same generator, particularly in non-deterministic and highly diverse generators like random regular and Erdős-Renyi. The differences between graphs of different generators is large enough that an MPNN can capture them. However, as we see the predictions gravitating towards the mean pattern of a generator, evidenced by the tighter band between the 2.5% and 97.5% percentile, distinguishing between graphs within each generator seems to be a task that MPNNs cannot generally perform. In these cases, the model tends to predict a less diverse set of patterns, leading to tighter percentile bands compared to reality (result also observable through Table 3 in Appendix C).

Table 1: Number of graphs with significance-profiles deemed "correct". Italic-underlined are for SAGE and the others GIN. Generators with $\geq$50%, $\geq$70% and $\geq$90% are highlighted. Error in non-deterministic graphs for SAGE is 15.0%, 15.2% for GIN and 11.4% for deterministic ones.

| Generator | "Correct" | | "Incorrect" | | Generator | "Correct" | "Incorrect" |
|---|---|---|---|---|---|---|---|
| Geometric-3D DD Graph | *035* | 224 | *314* | 125 | Balanced Tree | 191 | 129 |
| Duplication Divergence Graph | *280* | 303 | *069* | 046 | Barbell Graph | 185 | 135 |
| Extended Barabasi Albert Graph | *152* | 056 | *197* | 293 | Binomial Tree | 197 | 123 |
| Erdős-Renyi | *000* | 000 | *349* | 349 | Chordal Cycle Graph | 076 | 244 |
| Forest Fire | *209* | 119 | *140* | 230 | Circular Ladder Graph | 158 | 162 |
| Gaussian Random Partition Graph | *000* | 003 | *349* | 346 | Dorogovtsev Goltsev Mendes Graph | 107 | 213 |
| Random Limited Geometric Graph | *006* | 085 | *343* | 264 | Full Rary Tree | 244 | 076 |
| Newman Watts Strogatz Graph | *000* | 189 | *349* | 160 | Square Lattice | 167 | 153 |
| Powerlaw Cluster Graph | *336* | 301 | *013* | 048 | Hexagonal Lattice Graph | 156 | 164 |
| Random Regular Graph | *000* | 000 | *349* | 349 | Lollipop Graph | 085 | 235 |
| Watts Strogatz Graph | *000* | 031 | *349* | 318 | Star Graph | 106 | 052 |
| | | | | | Triangular Lattice Graph | 260 | 060 |
| Total | *1018* | 1311 | *2821* | 2528 | Total | 1932 | 1746 |

Assuming that the choice of null model had little impact in the difficulty of the problem, the conclusion of the ability of a model with expressivity at most equal to the 1-WL to be able to distinguish graphs of different generators can be seen as a partial empirical confirmation of an old result by Babai & Kucera (1979); Babai et al. (1980), regarding the 1-WL test being able to distinguish any random graph with high probability as the size of graph approaches infinity. Similarly, the apparent good performance of the tree generators is also theoretically backed by findings in Arvind et al. (2015). The conclusion of the inability of the model to distinguish graphs with high granularity among those in the same generator has theoretical backing for the case of the random regular generator (Babai & Kucera, 1979; Cai et al., 1989; Babai et al., 1980). As for the other generators, following the result in Babai & Kucera (1979); Babai et al. (1980), theoretically, it should be highly probable that a model as powerful as the 1-WL could distinguish most of the graphs, specially random ones, at inter and intra-generator level. However, according to our findings, this is not exactly true in practice, either because the model could not reach the 1-WL expressivity due to failing to approximate an injective function or because the bound does not work well in practical scenarios.

The model may be capable of accurately making inter-generator predictions by predicting a significance-profile corresponding to the mean of each generator, along with additional predictions gravitating around it. Following the formulation from Section 6, the "correct" predictions using the test set error will be evaluated assuming an error between 15.4% and 11.4% of the total error.

According to Table 1, more than 50% of the predictions for the deterministic dataset are satisfactory/correct. Not counting the regular graphs, known to not be distinguishable by 1-WL, SAGE got satisfactory predictions for 29.2% of the graphs and GIN for 37.6%. The generators whose graphs had more satisfactory predictions were those exhibiting a stronger mean pattern. In these cases, a model reaps significant gains from simply following the mean pattern. For instance, if we do not count the three problematic generators, regular graph, Gaussian random partition, and Erdős-Renyi, GIN has a satisfactory prediction rate of 46.9% and SAGE 36.5%. The worst-performing model achieves 29.2%, more than 80 times better than $T$, even with its lenient margin for accuracy.

**Answering question (2).** Yes, the model is learning the synthetic dataset (further reinforced by Figure 8). The model learns in two scales: inter-generator and intra-generator. It presents a good general capacity for inter-generator learning, meaning it does a fine job of distinguishing what is the generator of a graph. As for its intra-generator performance, it has a reasonable discriminative power. In the best case, for the non-deterministic segment, assuming 15.4% margin of error and a guaranteed signal match, it guesses correctly the significance-profile 46.9% of the times. As for the deterministic segment it predicts correctly 52.6% of the times, with a margin of error of 11.4%.

## 7.4 Model Predictions in the Real-World Dataset

Analysing the multiple figures from Section D.2, it becomes apparent that the inter-category (or inter-generator, in the context of the synthetic dataset) performance is far from optimal. The models exhibit identical predictions across graphs from different categories, even when these categories have distinct significance-profiles. Intra-category performance is even poorer than inter-category, with models like SAGE generating patterns that are too similar across graphs, though just distinct enough to avoid being captured by the persistence measures. This phenomenon is exemplified by

the predictions for the small biological category and the small interaction category observed in the SAGE model, by the medium-large interaction and medium-large social communication for GIN non-deterministic, among others. In the real-world data analysis, the SAGE model had the best performance with $12.5\%$ satisfactory predictions, seven correct in the small dataset. Other models had three or fewer. This result is $2.4\times$ worse than the poorest in the synthetic dataset, despite a $24.1\%$ margin of error, $\approx 1.6\times$ larger.

Interestingly, the models seem to have induced their own groupings based on similarities with the synthetic datasets used for training. For instance, real-world graphs like *ia-escorts-dynamic*, *coauthor-CS*, and *ia-primary-school-proximity* exhibited patterns resembling synthetic models like duplication-divergence, forest-fire, and geometric graphs, respectively. This suggests that the model predicts based on how similar a real-world graph is to the synthetic ones it has seen. While the model struggles to produce satisfactory predictions for real-world networks, it could help identify which synthetic model a real-world network most closely resembles. The closer the predicted pattern is to the true pattern, the more alike the synthetic and real networks.

**Points of divergence.** In network similarity discovery based on significance-profiles, two key concepts are crucial. First, the model's ability to distinguish networks is constrained by the expressivity of the space of the significance-profile used. The more expressive the space, the more reliable the ability of the model to distinguish networks. Secondly, if the model predicts similar profiles for two graphs $\mathcal{G}$ and $\mathcal{H}$, indicating they resemble a graph $\mathcal{F}$, this suggests $\mathcal{G}$ and $\mathcal{H}$ may originate from a process similar to $\mathcal{F}$. However, this conclusion is valid only if the true profiles of $\mathcal{G}$ and $\mathcal{H}$ are indeed similar; otherwise, the model's lack of expressivity leads to incorrect conclusions.

**Closeness to Random Weights.** A network with random weights can take any possible network as a value. However, achieving performance near the score of the trained network purely by chance is unlikely due to the complexity of the model. To address this, we propose that while the solution space for random models may differ, its mean score should be similar to that of the solution space of the trained model. Specifically, (1) models with random weights resemble those with high dropout, showing a tendency towards highly persistent patterns; (2) the trained model produces meaningful patterns for synthetic data; (3) the predictions of the model can significantly differ from true significance-profiles due to its limited expressivity. Thus, while a trained model predicts meaningful patterns ineffectively, a random one only predicts highly persistent patterns, leading to a similar average scores.

**Answering question (3).** The model learns from the synthetic dataset. The discrepancy comes from the synthetic data not accurately reflecting the real-world, at least when used by models limited by the 1-WL. Thus, we can only confirm that the knowledge extracted from the synthetic dataset by the used models is not enough to describe real-world data.

## 7.5 Validation of the Assumptions Made

Multi-target regression improves predictive accuracy for most graphs, except for graphs 4-clique and 4-path, likely due to limited benefit from shared information. Predicting significance profiles directly enhances performance for graphs with significant variation in subgraph occurrences, like the 3-path, triangle, 4-path, and 4-clique. Despite minor error increases in some cases, the overall gains in accuracy and computational efficiency make this approach preferable for motif estimation in the given null model. Section D.1 has a detailed exploration of the mentioned results.

## 8 Conclusions

Despite the lack of a GNN-based method specifically designed for predicting significance profiles, our current MPNN models combined with synthetic data are still deemed insufficient for real-world applications related to significance-profile discovery. The best performance benchmark shows a prediction accuracy of $12.5\%$ for small networks and $10.41\%$ for medium-large networks, each with a $24.1\%$ margin of error. In contrast, having into account their simplicity, the used models together with the presented formulation, for synthetic data, achieve good results, with a benchmark accuracy of $46.9\%$ with a $15.4\%$ margin of error for non-deterministic generators and $52.6\%$ with an $11.4\%$ margin of error for deterministic generators. This suggests that the models are promising for network categorisation by effectively distinguishing high-level differences between graphs.

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

# A   DATA DETAILS

## A.1   NON-DETERMINISTIC GENERATORS

For the Erdős-Renyi model Erdős & Rényi (1960), we mainly aim at creating graphs in the three (out of four) main topological phases a graph can be (Barabási & Pósfai, 2017). We exclusively uniformly control the number of nodes within each of the delineated phases, namely, the "critical", "supercritical" and "connected" states. This strategic regulation facilitates substantial variability in graph size while preventing an excessive escalation of the referred metric that could possibly impede further computational processing.

For the Watts-Strogatz (Watts & Strogatz, 1998) and Newman Watts-Strogatz (Newman & Watts, 1999), we regulated the generation based on the total number of nodes, the initial number of neighbours and the probability of rewiring in order to generate networks that represented key sections of the characterisation based on the clustering coefficient and path length, as given in Watts & Strogatz (1998).

For the extended Barabási-Albert model (Albert & Barabási, 2000), we defined as hyperparameters the total number of nodes and amount of connections a new node gains. Subsequently, we employ the equations delineated in the original article, values for the probabilities associated with the formation of new links ($p$) and the rewiring of existing connections ($q$) are derived. These computations aim to yield graphs characterised by a power-law degree distribution with an exponent ranging uniformly between 2 and 3.

For the cluster power-law (Holme & Kim, 2002), we vary uniformly the number of nodes and calculate the necessary probability according to the original study to obtain a clustering coefficient of 0.35, 0.45 or 0.55.

The duplication-divergence generator (Ispolatov et al., 2005) operates by randomly selecting a node $v$ from an initial graph and duplicating all edges connected to $v$ with a retention probability denoted as $\sigma$. Two of the selected regimes exhibit self-averaging behaviour concerning the number of edges, specifically when $0 < \sigma < e^{-1}$ and $e^{-1} < \sigma < 1/2$. The non-self-averaging regime is characterised by $1/2 < \sigma < 1$. More characteristics regarding the generated graphs, for example, the degree distribution, can be found in the original paper.

In the Gaussian random partition model, proposed by Brandes et al. (2003), $k$ groups of nodes are generated with $t$ nodes derived from a Gaussian distribution with mean $s$ and variance $v$. The connectivity between nodes in a group is given by a probability $p$, and the connectivity inter-groups is given by $q$. In this generator, we parameterise the number of nodes $|V|$, the size of the $k$ groups and the maximum number of allowed edges $|E_{\max}|$. Both the $p$ and $q$ probabilities are calculated to not exceed the maximum number of edges according to Equation 1.

$$q \leq min\left(1, \ \frac{2|E_{\max}|}{|V|^2 + |V|\left(\kappa \cdot s^{1/2} - s(\kappa + 1)\right)}\right) \tag{1}$$

$$p \leq min(1, \ \kappa \cdot q) \tag{2}$$

We defined $p$ as always having the possibility of going above $q$ because we would like to have networks that can have a some community structure in order to have a more diverse set of graphs. Hence, we put the upper bound of $p$ as being scaled over $q$ by $\kappa$, which we called over-attractiveness. The values used for the $v$ and $\kappa$ are 10 and 5 respectively. All other parameters are uniformly sampled from a predefined range[1].

In the case of the forest-fire model (Leskovec et al., 2007), we varied the number of nodes and the backward and forward probability between 0 and 0.4 (inclusive) to try to steer away from very aggressive Densification Power Law exponents and clique-like graphs, characteristics that, if severe, can hinder the subsequent steps from a computational point of view. With the values for the probabilities defined above, we expect to observe sparse networks that slowly "densify over time", together with decreasing diameter. All the graphs are made undirected after being generated.

---

[1]Details for the parameters available in the supplemental material.

For the random geometric graph, since some properties of the graph related to its connectedness, such as maximum cluster size and coefficient vary with the dimension of the unit hypercube used (Dall & Christensen, 2002; Penrose, 2003), we decided to vary the number of nodes and the dimension of the hypercube used between 2 and 5. However, we did not efficiently explore all possible configurations within the referred dimensions because we limited the number of edges. Similarly to a random geometric model, we used a random geometric model in 3D with duplication divergence (Higham et al., 2008). For this model, we followed Silva et al. (2023).

The last model in the non-deterministic segment is the random regular generator. In this case, the parameters subject to uniform variation were the total number of nodes and the degree assigned to each node, which once determined, remain constant across all nodes.

## A.2 DETERMINISTIC GENERATORS

We complemented the graphs generated by the non-deterministic generators with smaller amounts of graphs from deterministic generators. These generators have their network completely and without randomness determined once their parameters are chosen.

The first group of deterministic generators consists of multiple types of trees. We use the binomial tree model parameterised on its order and the balanced tree (full rary-tree) parameterised on its height and branching factor.

The second group is based on modified cycles. We use the circular ladder generator, varying the complete size of the graph and the chordal cycle (Lubotzky, 1994), also varying its complete size.

The third group is based on complete graphs and encompasses the barbell and lollipop graphs. The barbell graph is made of two complete graphs of size $k$ connected by a path of size $m$. The lollipop is a barbell graph with only one complete graph and the path. In order to not complicate subsequent steps, we carefully limited the size of the complete graphs.

The fourth group consists of the Dorogovtsev-Goltsev-Mendes model (Dorogovtsev et al., 2002). This generator modulates a scale-free discrete degree distribution with exponent $1 + ln\,3/ln\,2$ by using a rather simple rule: "At each time step, to every edge of the graph, a new vertex is added, which is attached to both the end vertices of the edge." (in Dorogovtsev et al. (2002)). We vary the magnitude of the number of nodes and edges by changing $n$, resulting in $3(3^n + 1)/2$ and $3^{n+1}$ nodes and edges respectively.

The fifth group consists of lattices. Namely, we use 2D hexagonal, triangular lattices and 3D square lattices. The first 2 lattices have the option of allowing for boundary periodicity. All lattices vary in terms of the size of each dimension.

Finally, the last group consists of star graphs of various sizes.

Since the types of graphs that the deterministic generators generate are not subject to randomness, it is redundant to create multiple graphs for each set of parameters. However, in order to introduce a degree of randomness to the deterministic graphs, we introduced a probability of random rewiring of a percentage of edges after the graph is generated. The rewiring procedure for a single edge consists of selecting an edge $(u, v)$ from a graph $\mathcal{G}$, deleting it and attaching one of the ends, $u$ or $v$, to another node $w$. If $u$ is picked and $(u, w)$ already exists, then $\mathcal{G}$ will exit the procedure with one less edge. Since we want some variability but still want to preserve the original deterministic graphs, for each generator, two sets of graphs $\mathbb{S}_1$ and $\mathbb{S}_2$ will be generated where each set goes through all the proposed generator parameters. After that, $\mathbb{S}_1$ is not subject to any rewiring, and for each graph in $\mathbb{S}_2$, $p\%$ of its edges are rewired according to the procedure described earlier. According to this methodology, we generated four versions. The first had 2 edges swapped, the second 25% of the edges swapped, the third 10% and the fourth 60%. We stick to version two due to being the best performing one according to preliminary tests. This fact means that 25% seems to be a good choice of random-rewiring so that the information encoded in the deterministic graphs is maximised.

## A.3 EXACT DATASET PARTITION

Initially, we will conduct separate experiments using the two segments of the produced data, the deterministic and the non-deterministic. Furthermore, the split in train-validation-test is stratified

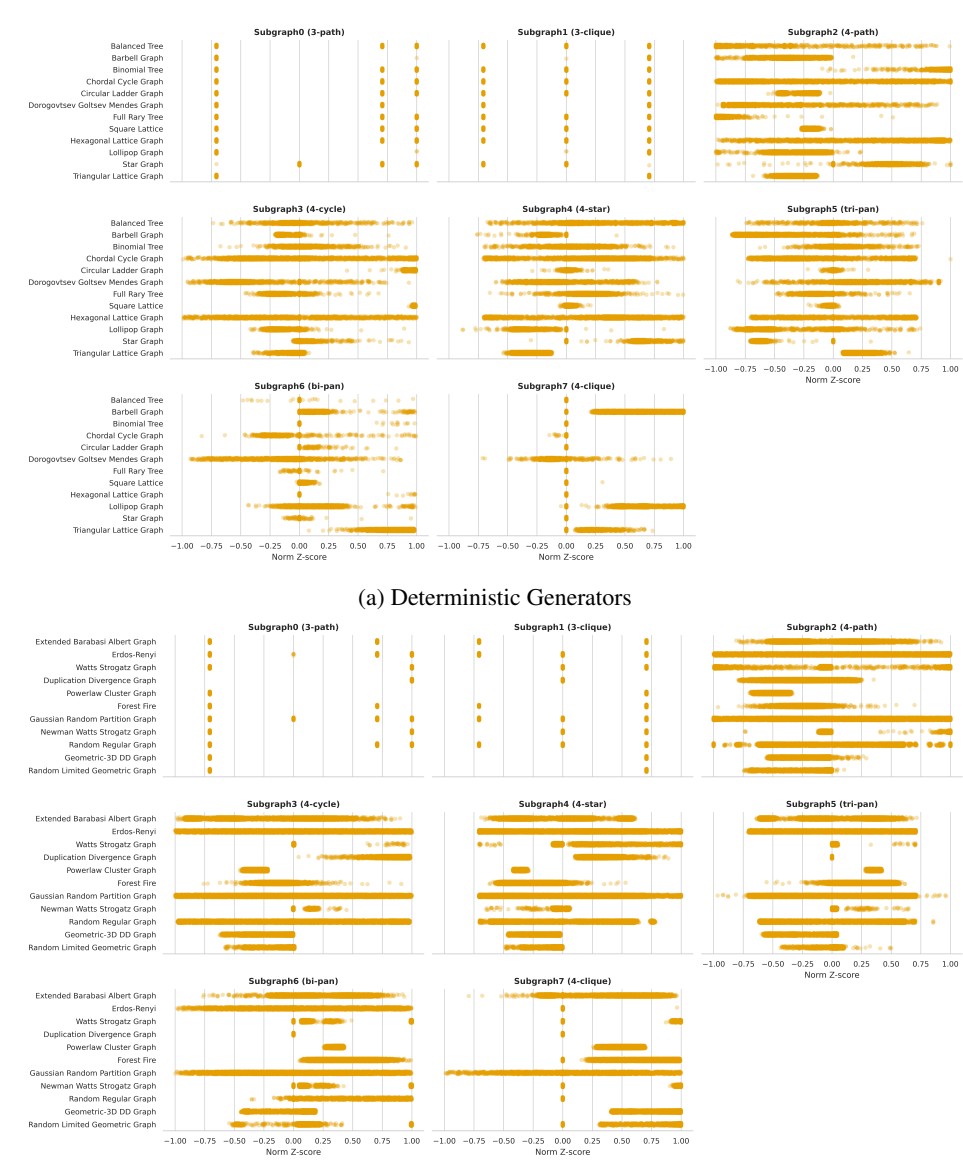

Figure 2: Distribution of the values the significance profile of each graph in $\Omega$ takes in the synthetic dataset. Each point represents a graph.

by random sampling a percentage $p$ of each of the generators for each segment. In the case of the deterministic segment, we use all 3200 graphs available. With $p = 0.7$, the training set has $3200 \times 0.7 \times 12 = 26880$ graphs, and the validation set has $3200 \times 0.2 \times 12 = 7680$ graphs. The remaining 10% are used for the test set. We avoided using larger datasets due to memory restrictions. As for the non-deterministic segment, in order for it to have a comparable total size to the deterministic segment, we sampled $3490 \times 0.7 \times 11 = 26873$ graphs and the validation set $3490 \times 0.2 \times 11 = 7678$.

## A.4 PATTERN INTERCONNECTIVITY

Figure 2 adopts a view of individual cases at the cost of a global view of the patterns. Each dot corresponds to an individual graph. Each point is slightly transparent to attempt to give the notion of density.

Following the discovery introduced in Section 5 regarding the symmetry of the score of size three patterns, we have a strong indication that even without the normalisation of the Z-score, the number of occurrences between connected graphs of the same size is highly related. More formally, the relation between the Z-scores of the graphs of size three can be described as follows. Let x be a random variable denoting the number of induced occurrences of triangles in any graph that follows the degree distribution $D$. Let y be a random variable denoting the occurrence of induced 3-paths in any graph that follows the degree distribution $D$. Equation 3 gives the relation between Z-scores of x and y.

$$X = \begin{cases} 0, & \text{if} \quad \text{y} - \mu_\text{y} = 0 \\ -\frac{\sigma_\text{x}}{\sigma_\text{y}}(Y - \mu_\text{y}) + \mu_\text{x}, & \text{otherwise} \end{cases} \tag{3}$$

When standardised to a mean of $0$ and a standard deviation of $1$, the Z-scores of both variables, exhibit symmetry. Hence, it is possible to express their non-standardised values as linear combinations of each other. Considering the mean and standard deviation of the counts of 3-paths and triangles for $D$, given any graph $\mathcal{G}$ that follows $D$, it is possible to get the concrete count of triangles from the count of 3-paths and vice-versa.

Even though from a practical point of view, the result from Equation 3 has little implications due to the dependence on the first raw moment and the second central moment of both the distributions of x and y, it presents a strong indication of what was postulated in Section 4 regarding the connectivity between the graphs selected for $\Omega$. In this case, the relation is so strong that we believe to be redundant to try to predict both scores. Moreover, following the normalisation procedure, the restrained nature of the result raises questions about the choice of modelling the problem as a regression task for the size three graphs. However, despite these observations, we stick to our initial formulation since in theory it does not undermine the ability of the model.

The result experimentally verified in the above paragraphs can be seen as a small extension of Ginoza & Mugler (2010) and Wegner (2014) to undirected patterns of size three. In particular, adapting from Wegner, Equation 4 displays the conservation law for the number of induced 3-paths.

$$\text{\#3-paths}_{ind} = \underbrace{\text{\#3-paths}_{\overline{ind}}}_{\text{fully defined by degree sequence}} - \underbrace{3\text{\#triangles}}_{\text{not fully defined by degree sequence}} \tag{4}$$

Since the number of non-induced 3-paths depends only on the degree sequence $\left(\sum_{i=0}^{|V|}\binom{|N(i)|}{2}\right)$, it will not change under the configuration model. Hence, the number of induced 3-paths is a variable that once the degree sequence is fixed, depends only on the number of triangles. As for the number of triangles, they depend on the order the edges are added to the graph under equation 5.

$$\text{total triangles} = \sum_{t=0}^{|E|} \text{\#triangles}_t \tag{5a}$$

$$\text{total 3-path} = \sum_{t=0}^{|E|} (\text{\#3-path}_t - \text{\#triangles}_t) \tag{5b}$$

$$\text{\#triangles}_{t+1} = |\{w | w \in N(u^t) \wedge w \in N(v^t)\}| \tag{5c}$$

$$\text{\#3-path}_{t+1} = |N(u^t)| + |N(v^t)| - 2\text{\#triangles}_{t+1} \tag{5d}$$

where nodes $u$ and $v$ represent the nodes that were connected by an edge at iteration $t$. Hence, any realisation of a degree sequence through the configuration model will always have its number of induced 3-paths negatively correlated with the number of triangles.

Regarding the relation between graphs of size three and graphs of size four, by analysing Figure 3, it is possible to understand that there is a relation between the significance profiles of these graphs. This relationship is particularly pronounced concerning the 4-star, tri-pan and 4-clique, as the values

of the significance profiles assumed by these graphs are distributed across the spectrum centred at 0, contingent upon the value held by the 3-path. As for the 4-cycle and bi-fan, this relation is not as strong. For the 4-cycle, we learn that the values are mostly zero when the significance-profile for the 3-path is negative and is quite dispersed across the space otherwise. As for the bi-fan, even though hard to discern from the figure, $46.2\%$ of the values are $0$ when the significance-profile for the 3-path is positive, and $69.7\%$ are between $-0.1$ and $0.1$ for the same conditions.

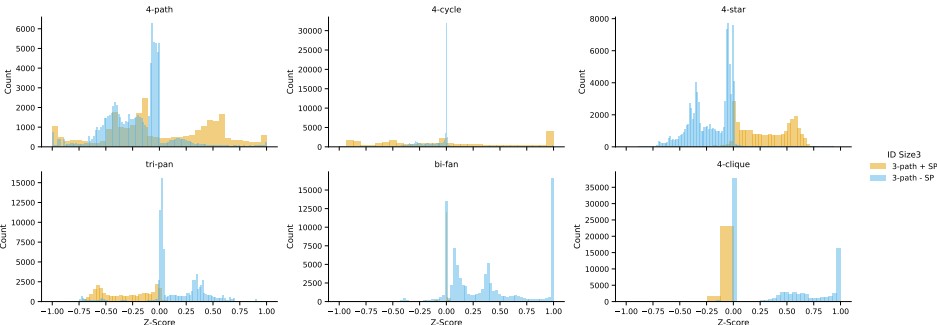

Figure 3: Distribution of the significance profiles for the graphs of size 4, given the value the 3-path took. The positive value corresponds to $0.707106$ and the negative to $-0.707106$.

## A.5 REAL-WORLD DATASET

The selected type categories are described in list A.5. The numbers between the square brackets in each bullet point correspond to the number of networks each category has in each of the scale categories from the smallest to the largest scale.

- **ANIMAL SOCIAL**: [10/8] networks of the social behaviour of non-human animals incorporating a spectrum that includes ants, dolphins, lizards, sheep, and other examples.

- **BIOLOGICAL**: [10/10] networks of protein-protein interactions, a metabolic network of small organisms and a network of connections between diseases in humans by the number of shared genes.

- **BRAIN**: [9/10] networks of connectomes of diverse regions such as the cerebral cortex, interareal cortical network, and neural synaptic, among others, of multiple species such as cats, worms, mice, macaques and humans.

- **CHEMOINFORMATICS**: [10/0] networks of multiple different enzyme structures.

- **COLLABORATION CITATION** [6/8]: networks of citations of papers and collaborations between authors.

- **INFRASTRUCTURE**: [5/7] Electric grids and road networks.

- **INTERACTION**: [5/6] Networks of physical contact between humans in various contexts, together with some digital contact, for example, by e-mail or a phone call.

- **SOCIAL COMMUNICATION**: [2/10] Interaction between humans in social networks such as mutually liked Facebook pages, friendship connections and retweets.

Figure 4a and Figure 4b show a summary of the number of edges and nodes for the different type and scale categories. The red dashed lines represent the average of the minimum node (or edges) quantity, the average of the mean node (or edges) quantity and the average of the maximum node (or edges) quantity respectively, calculated for all 23 graph models used in the synthetic dataset.

In Figure 4a and Figure 4b, a noticeable distinction is observable in the delineation of scale categories based on the type category. The distinction between scale categories is influenced by the type category of the network. For example, ANIMAL SOCIAL networks tend to be smaller than INFRASTRUCTURE networks in the medium-large category. However, this relationship varies by scale, as both network types exhibit similar sizes in the small-scale category, contrasting with their

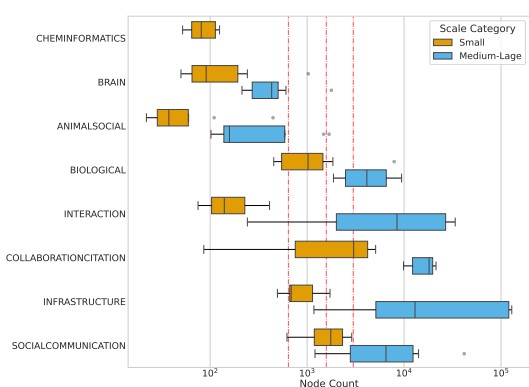

(a) Summary of how the number of nodes is distributed for the real networks.

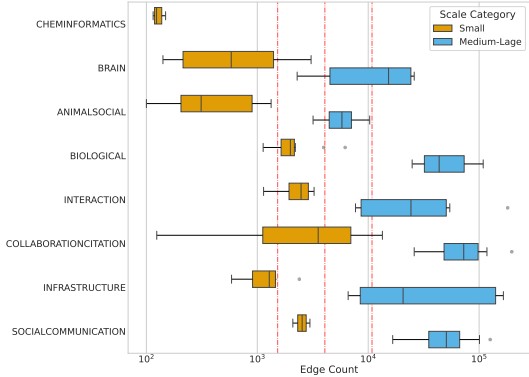

(b) Summary of how the number of edges is distributed for the real networks.

Figure 4: Summary of the distribution of the node and edge count of the real networks. All data is presented in logarithmic scale.

medium-large behaviour. In any case, in the general scenario, we expect that the small-scale category encloses networks from being smaller than an average network from the training set until networks that can be slightly larger than the mean training network. As for the medium-large category, they hold networks that can be around the average size of a training network to networks that are several orders of magnitude larger than the average size of a training network. The dashed red lines in Figure 4a and Figure 4b help validate this statement.

The supplemental details have further details regarding the real-networks used. In summary, we have 56 networks in the small-scale category and 59 in the medium-large category. If a graph in the test set was not already a simple undirected static graph when it was obtained, we transformed it in a graph following said conditions.

## B  INITIAL BASE MODEL

The model ($\mathcal{B}$) consists of three modules. The first module consists of $K$ layers of a GNN. The job of the first module is to work on the graph data and adjust the node embeddings so that the second module, a global pooling function, can summarise them into a single graph-level embedding. The third module is an MLP that takes as input the graph embedding and will adapt it to output the final prediction for the normalised Z-scores of the graphs in $s$. Figure 5 shows a diagram of the model.

All the optimisations for the hyperparameters of $\mathcal{B}$ will be performed by Optuna (Akiba et al., 2019) with 450 rounds of suggestions of hyperparameters, orchestrated through Ray (Liaw et al., 2018; Moritz et al., 2018). Moreover, the hyperparameter sampling procedure employed the Tree-Structured Parzen Estimator (Watanabe, 2023), while the pruning strategy was executed through the

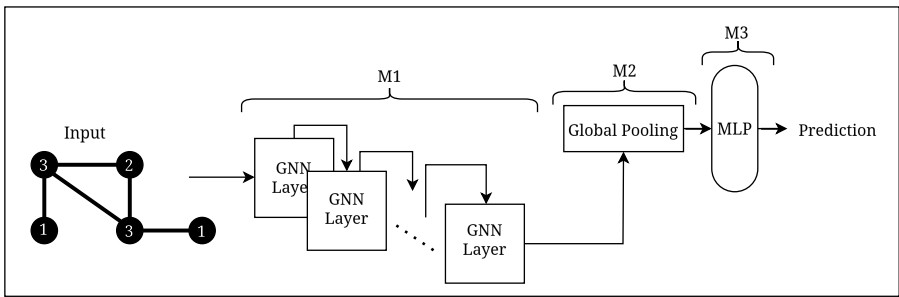

Figure 5: Illustration of the base model $\mathcal{B}$ divided in three modules, M1, M2 and M3.

Table 2: Break down of the hyperparameter space used for $\mathcal{B}$.

| | Min | Max | | Epochs | Batch Size | Learning Rate (log) |
|---|---|---|---|---|---|---|
| M1-GNN Depth | 2 | 3 | | | | |
| M1-Hidden Dimension | 6 | 16 | | | | |
| M1-GNN Dropout | 0.0 | 0.9 | | | | |
| M1-Jumping Knowledge | max, cat, lstm | | | 100 | 16, 32, 64, 128, 256 | [0.00001, 0.001] |
| M2-Global Pool | add | | | | | |
| M3-MLP Depth | 2 | 6 | | | | |
| M3-Hidden Dimension | 6 | 16 | | | | |
| M3-MLP Dropout | 0.2 | 0.9 | | | | |

application of the median rule (Golovin et al., 2017). Table 2 presents the hyperparameter space used for model $\mathcal{B}$.

The asymmetry in the hyperparameter space presented in Table 2 stems from our choice of preemptively test a slightly larger hyperparameter space and identify some values that resulted in very bad results. From this early testing phase, we also narrowed down M3 from a global add, mean or max function to just the global add function. This result aligns with some limitations that are known for the mean and max pooling functions (Xu et al., 2019). Furthermore, the fixed values of 100 epochs can be shortened not only by the pruner but also by an early-stopping module with a grace period of 25 epochs, synced with the median pruner, and patience of other 25 epochs of not seeing an improvement for the global minimum loss. Moreover, since we believe our problem does not need very long range dependencies since the structures in $\Omega$ can be fully defined by a hop size of 2, in order to try to limit the problem of over-smoothing we limited the maximum number of GNN layers to 3 based on the findings that most networks have a small diameter (Albert et al., 1999; Barabási et al., 2000; Watts & Strogatz, 1998). By limiting the GNN layers, we also hope to reduce over-squashing.

For the initial tests, we performed several experiments on the synthetic dataset described in Section 5 for different M1 modules. Each experimental iteration comprised 450 trials, each uniquely characterised by a distinct combination of hyperparameters suggested by Optuna.

The four different GNNs used were GAT (Veličković et al., 2018), SAGE (max-pooling) (Hamilton et al., 2017), GCN (Kipf & Welling, 2017) and GIN (Xu et al., 2019) and are all 1-WL limited. Most of the training was done using a single NVIDIA RTX 3090 and later a NVIDIA RTX A6000.

## C  METHODOLOGIES

**Persistent Patterns.**   To evaluate the impact of a value $t$ in the number of persistent patterns, we employ agglomerative clustering with complete linkage over the significance profiles with $t$ as a stopping point for agglomeration. Using the complete linkage over $d_\infty$ ensures that all significance profiles in a cluster remain within $t$ of each other, effectively counting the number of persistent patterns through the number of clusters and their persistency through the size of each cluster. To discover $t$, we test different values and iteratively evaluate their impact on the quantity of patterns induced. The selected value is the lowest one that produces the largest drop in the number of per-

Table 3: Summary of the number of persistent patterns and their persistency. The number of clusters gives the number of persistent patterns and the statistics about their size pertain to $\rho_s$.

| Segment | Model | Q1 | Q2 | Q3 | Min | Mean | Max | STD | Number Clusters | Number of SPs | Threshold |
|---|---|---|---|---|---|---|---|---|---|---|---|
| ND - Test | True | 1.00 | 1.00 | 1.00 | 1.00 | 1.65 | 56.00 | 3.15 | 2325 | | |
| | GIN | 1.00 | 1.00 | 2.00 | 1.00 | 19.39 | 710.00 | 73.92 | 198 | 3839 | 0.01 |
| | SAGE | 1.00 | 1.00 | 2.00 | 1.00 | 11.53 | 751.00 | 63.79 | 333 | | |
| D - Test | True | 1.00 | 1.00 | 1.00 | 1.00 | 1.90 | 38.00 | 3.25 | 1935 | 3678 | 0.01 |
| | GIN | 1.00 | 1.00 | 2.00 | 1.00 | 4.73 | 520.00 | 25.88 | 777 | | |
| Medium-Large | True | 1.00 | 1.00 | 1.00 | 1.00 | 1.38 | 5.00 | 0.94 | 42 | | |
| | GIN-ND | 1.25 | 2.00 | 5.25 | 1.00 | 4.14 | 20.00 | 5.07 | 14 | 58 | 0.08 |
| | SAGE-ND | 1.00 | 1.00 | 3.00 | 1.00 | 2.15 | 6.00 | 1.61 | 27 | | |
| | GIN-D | 1.00 | 1.00 | 2.00 | 1.00 | 2.90 | 27.00 | 5.86 | 20 | | |
| Small | True | 1.00 | 1.00 | 2.00 | 1.00 | 1.51 | 7.00 | 1.19 | 37 | | |
| | GIN-ND | 1.00 | 1.00 | 3.00 | 1.00 | 2.95 | 12.00 | 3.41 | 19 | 56 | 0.21 |
| | SAGE-ND | 1.00 | 2.00 | 3.50 | 1.00 | 2.95 | 10.00 | 2.57 | 19 | | |
| | GIN-D | 1.00 | 1.00 | 3.00 | 1.00 | 2.67 | 14.00 | 3.43 | 21 | | |

sistent patterns (similar to the elbow method for clustering). Table 3 conveys summary statistics regarding the discovered persistency of patterns.

**Red Line.** Equation 6 delineates the derivation of the expected loss for the baseline depicted as a red line. The first step in said equation comes from expanding the square and the second step from the definition of variance for a standard uniform distribution. Conceptually, the red line embodies the unequivocal baseline for any model endeavouring to address the problem of predicting normalised Z-scores as delineated in Section 4. Any model falling short of the benchmark set by the red line can be confidently deemed inadequate.

$$\frac{1}{n} \sum_{i=1}^{n} \mathbb{E}\big((y_i - \hat{Y}_i)^2\big) \tag{6a}$$

$$\equiv \frac{1}{n} \sum_{i=1}^{n} \big(y_i^2 + Var(\hat{Y}_i)\big) \tag{6b}$$

$$\equiv \frac{1}{n} \sum_{i=1}^{n} (y_i^2 + 1/3) \tag{6c}$$

$$\equiv 1/3 + 2/n \tag{6d}$$

**Blue Line.** The derivation is similar to the one employed for the red line, hence omitted. Compared to the red line, the blue line defines an improved standard that any model that tries to predict the normalised Z-score as postulated in Section 4 must also clear. To delineate the value of the blue line, we formulated a model enabling the stochastic prediction of values that adhere to the constraint that the sum of the squared values of the scores of the graphs of each group of $\Omega$ must be equal to 1. We chose to articulate the model using a set of independent standard normal variables $\mathbb{S} = \{x_1, \ldots x_n\}$, which are then normalised by the Euclidean norm of the respective $\mathbb{S}$. This particular formulation seems to manifest a good distribution across the space formed by $\mathbb{S}$ in terms of uniformity, especially when considering the increasingly improbable nature, attributed to both the concentration of measure phenomenon and Dvoretzky's theorem (Pisier, 1989; Giannopoulos & Milman, 2000), of truly attaining a random uniform distribution of points across the entire volume of a compact, symmetric, convex subset within an $n$-dimensional Banach space, like the $n$-Euclidean space, $S$, delineated by the process used to generate random guesses. On another note, precisely due to this observation, we conjecture that the problem of interest is ill-posed in very high dimensional spaces, meaning it becomes increasingly hard to have a meaningful significance-profile and thus predict them as the groups of $\Omega$ increase in cardinality.

**Light-Pink Bar.** To ascertain the referred mean error that the light pink bar depicts, we randomised the weights of each model a total of 100 times and predicted the significance-profiles for the real-world dataset for each randomisation. In conjunction with the mean error estimation, we provide an error bar indicative of one standard deviation.

# D EXPERIMENT RESULTS

Figure 6 and 7 shows the summary of the results from the 450 rounds of hyperparameter optimisation for each model used in M1. The solid line represents the mean score, and the semi-transparent bound around each line represents the standard error. The displayed metrics are the MSE for the train and validation data, the median absolute error, $med(\{med_i(|y_i - \hat{y}_i|, \forall i \in |\boldsymbol{y}|)\})$, the maximum absolute error calculated for a full prediction of a significance profile, and the mean value for the worst-performing prediction of a graph from $\Omega$. The maximum error is given by $max(\{\sum_{j \in |\Omega|} |\boldsymbol{Y}_{[i,j]} - \hat{\boldsymbol{Y}}_{[i,j]}|, \forall i \in |D_{\text{valid}}|\})$ where $\boldsymbol{Y}$ is a 2-d matrix where the first dimension gives the number of examples in the validation dataset and the second dimension the length of $\boldsymbol{s}$. As for the mean value of the worst-performing predictions, it is given by $mean(\,max\{\sum_{i \in |D_{\text{valid}}|} |(\boldsymbol{Y}_{[i,j]} - \hat{\boldsymbol{Y}}_{[i,j]})|, \forall j \in |\Omega|\}\,)$.

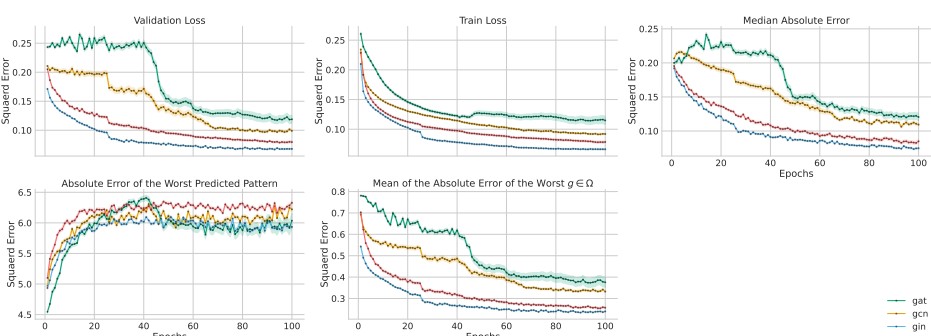

Figure 6: Learning curves for the various backends used for M1 when trained with the deterministic segment of graph generators.

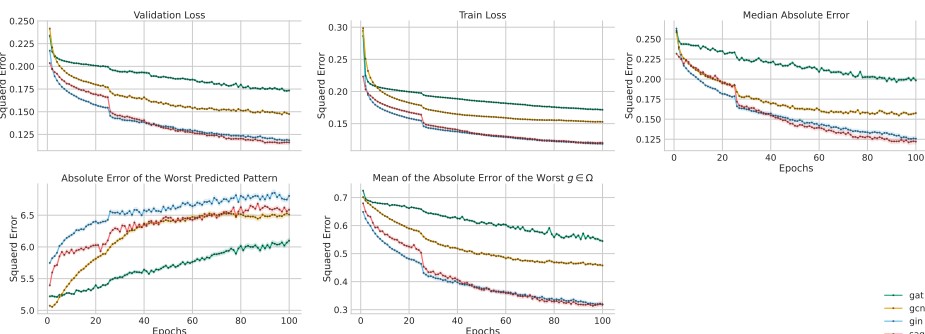

Figure 7: Learning curves for the various backends used for M1 when trained with the non-deterministic segment of graph generators.

The learning curves for the deterministic segment (Figure 6) show that all models improve significantly within the first 50 epochs, especially in all metrics except for the maximum absolute error. GIN outperforms all other models by a wide margin, prompting its selection for further analysis. For the non-deterministic segment (Figure 7), the performance of GraphSAGE and GIN is very close, with GraphSAGE holding a slight numerical edge. Since both models perform comparably, both will be retained for further evaluation.

## D.1 VALIDATION OF THE ASSUMPTIONS MADE

The two main assumptions by us proposed regards to using multi-target regression and directly predicting the significance-profiles of the chosen graphs.

**Single-target vs. Multi-target**. To allow this comparison, we trained eight models, each for one of the graphs in $\Omega$. The data used was the non-deterministic synthetic dataset. For the task, we utilised

Table 4: Percentiles int the validation set of the squared error and their percent decrease/increase comparing the multitarget to the single target model. Order: 3-path (0), triangle (1), 4-path (2), 4-cycle (3), 4-star (4), tailed-triangle (5), 4-chord-cycle (6), 4-clique (7).

| Graph | 7 | 7 | 6 | 6 | 5 | 5 | 4 | 4 | 3 | 3 | 2 | 2 | 1 | 1 | 0 | 0 |
|---|---|---|---|---|---|---|---|---|---|---|---|---|---|---|---|---|
| Type | single | multi | single | multi | single | multi | single | multi | single | multi | single | multi | single | multi | single | multi |
| 100% | 1.835 | 2.868 | 1.830 | 1.126 | 1.126 | 0.816 | 1.498 | 0.913 | 1.056 | 1.268 | 2.275 | 2.388 | 1.814 | 2.193 | 2.940 | 2.621 |
|  | +56.294% | | -38.470% | | -27.531% | | -39.052% | | +20.076% | | +4.967% | | +20.893% | | -10.850% | |
| 95% | 0.294 | 0.250 | 0.376 | 0.334 | 0.304 | 0.320 | 0.350 | 0.348 | 0.547 | 0.502 | 0.517 | 0.489 | 1.294 | 0.741 | 1.463 | 0.808 |
|  | -14.966% | | -11.170% | | +5.263% | | -0.571% | | -8.227% | | -5.416% | | -42.736% | | -44.771% | |
| 75% | 0.080 | 0.091 | 0.085 | 0.050 | 0.052 | 0.041 | 0.083 | 0.053 | 0.067 | 0.038 | 0.082 | 0.100 | 0.210 | 0.042 | 0.357 | 0.048 |
|  | +13.750% | | -41.176% | | -21.154% | | -36.145% | | -43.284% | | +21.851% | | -80.000% | | -86.555% | |
| 50% | 0.007 | 0.009 | 0.031 | 0.013 | 0.005 | 0.006 | 0.008 | 0.007 | 0.005 | 0.003 | 0.007 | 0.012 | 0.051 | 0.007 | 0.042 | 0.007 |
|  | +28.571% | | -58.065% | | +20.000% | | -12.500% | | -40.000% | | +71.429% | | -86.275% | | -83.333% | |
| 25% | 0.000 | 0.001 | 0.002 | 0.001 | 0.001 | 0.000 | 0.001 | 0.001 | 0.001 | 0.000 | 0.000 | 0.002 | 0.005 | 0.000 | 0.012 | 0.003 |
|  | +100.000% | | -50.000% | | -100.000% | | 0.00% | | -100.000% | | +100.000% | | -100.000% | | -75.000% | |

Table 5: Percentiles in the test set of the absolute difference between the true and predicted significance-profile by direct estimation (SP) and their percent decrease/increase comparing to the predictions using a multi-target model with graph frequencies as output. Order: 3-path (0), triangle (1), 4-path (2), 4-cycle (3), 4-star (4), tailed-triangle (5), 4-chord-cycle (6), 4-clique (7).

| Graph | 7 | 7 | 6 | 6 | 5 | 5 | 4 | 4 | 3 | 3 | 2 | 2 | 1 | 1 | 0 | 0 |
|---|---|---|---|---|---|---|---|---|---|---|---|---|---|---|---|---|
| Type | Count | SP | Count | SP | Count | SP | Count | SP | Count | SP | Count | SP | Count | SP | Count | SP |
| 75% | 0.806 | 0.222 | 0.248 | 0.323 | 0.150 | 0.199 | 0.120 | 0.227 | 0.073 | 0.173 | 0.541 | 0.298 | 0.362 | 0.172 | 0.412 | 0.278 |
|  | -72.429% | | +30.320% | | +33.280% | | +89.686% | | +137.311% | | -51.763% | | -68.122% | | -32.566% | |
| 50% | 0.161 | 0.116 | 0.046 | 0.109 | 0.065 | 0.056 | 0.065 | 0.083 | 0.042 | 0.083 | 0.202 | 0.126 | 0.476 | 0.072 | 0.335 | 0.079 |
|  | -28.068% | | +140.276% | | -14.035% | | +28.404% | | +99.939% | | -37.503% | | -84.891% | | -76.506% | |
| 25% | 0.081 | 0.044 | 0.025 | 0.042 | 0.016 | 0.036 | 0.019 | 0.031 | 0.018 | 0.020 | 0.083 | 0.041 | 0.396 | 0.027 | 0.231 | 0.021 |
|  | -46.267% | | +65.427% | | +127.724% | | +61.485% | | +9.853% | | -50.814% | | -93.296% | | -90.737% | |

the GIN variant of MPNNs, as it demonstrated both theoretical and practical superiority among available options. The models were trained without any prior assumptions; they were initialised with the same hyperparameter space as all other models, allowing the optimiser to explore the entire parameter space. Consequently, the models are designed to specialise in their respective graph. Table 4 shows the percentiles for the squared difference between the true and predicted (using GIN model from Figure 7) significance-profile in the validation dataset for each of the eight graphs of $\Omega$. Each percentile has a percent comparison between the results of the multiple models.

Following the results from Table 4, apart from graph 7 (four-node clique) and 2 (four-path), generally, all graphs show an improvement in the predicted score when using multi-target regression. This result confirms our expectation delineated in Section 4 that graphs that predicting together graphs that share common traits can improve the outcome. Regarding the increased error observed in graphs 7 and 2, we hypothesise that this may be due to their limited benefit from shared information. Other graphs do not possess enough encoded information to be leverage by the shared knowledge to outperform a specialised model. Regardless, having into account the magnitude of the increase/decrease of the the errors, we believe that multi-target is superior for motif estimation. One other added benefit is the need to train only a single model that maintains competitive training times comparing to single-target regression.

**Estimating Frequency vs. Estimating Significance-Profiles.** To allow this comparison, we trained a model to estimate the frequency of the graphs of $\Omega$ directly. The data used was the non-deterministic synthetic dataset. For the task, we utilised the GIN variant of MPNNs, as it demonstrated both theoretical and practical superiority among available options. The model was trained without any prior assumptions; it was initialised with the same hyperparameter space as all other models, allowing the optimiser to explore the entire parameter space. Consequently, the model is designed to specialise in frequency estimation. Table 5 shows the percentiles for the absolute difference between the true and predicted significance-profile, by either direct estimation (SP) or frequency estimation (Count) in the test dataset for each of the eight graphs of $\Omega$. Each percentile has a percentage comparison between the outcomes of the two models.

To circumvent the need to generate 500 random networks according to NULL for each network in the test set and subsequently estimate the frequency of each subgraph within them, we opted for an estimation of the predicted Z-Scores. Let x represent a random variable corresponding to the frequency of subgraphs in a given degree-distribution as estimated by the trained model. We decompose x and its associated Z-score calculation into two variables, y and z, as shown in Equation 7.

The variable y denotes the actual frequency of a subgraph, while z captures the error introduced by the model's approximation.

$$\hat{Z} = \frac{x - \mathbb{E}[\mathrm{x}]}{Var[\mathrm{x}]^{1/2}} \tag{7a}$$

$$\hat{Z} = \frac{(y + z) - \mathbb{E}[\mathrm{y} + \mathrm{z}]}{\left(Var(\mathrm{y})^2 + Var(\mathrm{z})^2\right)^{1/2}} \tag{7b}$$

Assuming that the difference between a value $z \sim \mathrm{z}$ and $\mu_{\mathrm{z}}$ is proportional to $\sigma_{\mathrm{z}}$, minding signal indetermination, Equation 7 originates Equation 8.

$$\hat{Z} = \frac{(y - \mathbb{E}[\mathrm{y}]) + (z - \mathbb{E}[\mathrm{z}])}{\left(Var(\mathrm{y})^2 + Var(\mathrm{z})^2\right)^{1/2}} \tag{8a}$$

$$\hat{Z} = \frac{(y - \mathbb{E}[\mathrm{y}]) \pm \sigma_{\mathrm{z}}}{\left(Var(\mathrm{y})^2 + Var(\mathrm{z})^2\right)^{1/2}} \tag{8b}$$

All values in Equation 8 are known since they were either acquired during the training of the model (values regarding z) or were collected during the dataset construction (values regarding y).

By approximating the Z-scores predicted by models that perform frequency estimation, and subsequently normalising them as described in Section 4, we are able to compare these approximations with the outputs of models that directly predict the normalised Z-scores (significance profiles). We employ the percentiles of the absolute differences between the true significance profiles and those directly estimated by the models, as well as between the true significance profiles and those estimated via Equation 8. Table 5 presents this comparison. The values under "Count" correspond to the minimum difference (hence worst case comparison) resulting of all valid signal combinations according to Equation 8b.

Following the results from Table 5, we conclude that, generally, predicting significance-profiles directly improves the scores for the 3-path (graph 0), triangle (graph 1), 4-path (graph 2) and the 4-clique (graph 7). All others present a general score that is worse when directly predicting the significance-profile. The graphs that experienced overall improvement are those characterised by significant variation in their number of occurrences, depending on the type and size of the network. This result likely arises from the model's enhanced ability to handle the extreme differences in the frequency of subgraph occurrences in networks with substantial size and topological disparities. Despite, the deteriorating the results for graphs 3 through 6, apart from 75% for graph 3 and 50% for graph 6, the increase does not magnify the order of magnitude of the errors significantly. Furthermore, the decrease for the other graphs is larger and more significant. Hence, we believe predicting significance-profiles directly to be an overall improvement to predicting graph frequencies, in the context of motif estimation for the chosen null model. Furthermore, predicting significance-profiles directly is much cheaper computationally from a motif calculation point of view.

Joining the result from Table 4 and 5, in the context of motif estimation, we believe that using multi-target regression to predict significance-profiles directly outperforms using models to perform subgraph frequency estimation using either single or multi-target regression.

## D.2  PREDICTIONS

Figures 8 display a summary of the predictions for each generator made by each selected model.

Regarding the real-world dataset, since the number of graphs per category is much smaller than in the synthetic dataset, basing an analysis solely on the mean profile and percentile band can be deceiving. Regardless, we make available such figure available (Figure 12 and 13). Conversely, that scarcity allows for a more comprehensive individual analysis. Still, the volume of graphs is to high to present all images. Figures 9a through 11d show some examples. To see all predictions follow the README in the supplemental material.

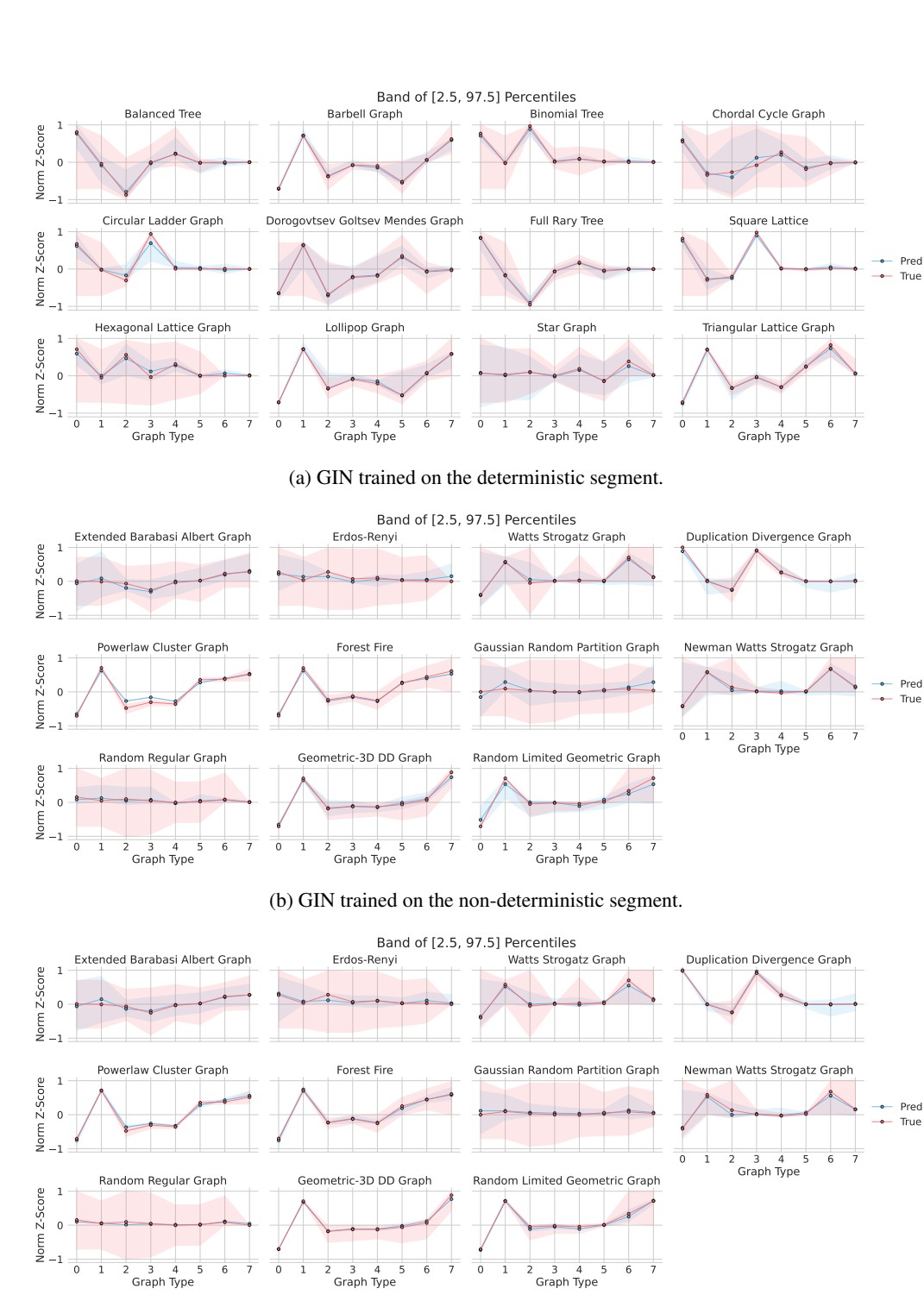

(a) GIN trained on the deterministic segment.

(b) GIN trained on the non-deterministic segment.

(c) SAGE trained on the non-deterministic segment.

Figure 8: Predictions for each model in each of their corresponding synthetic test datasets.

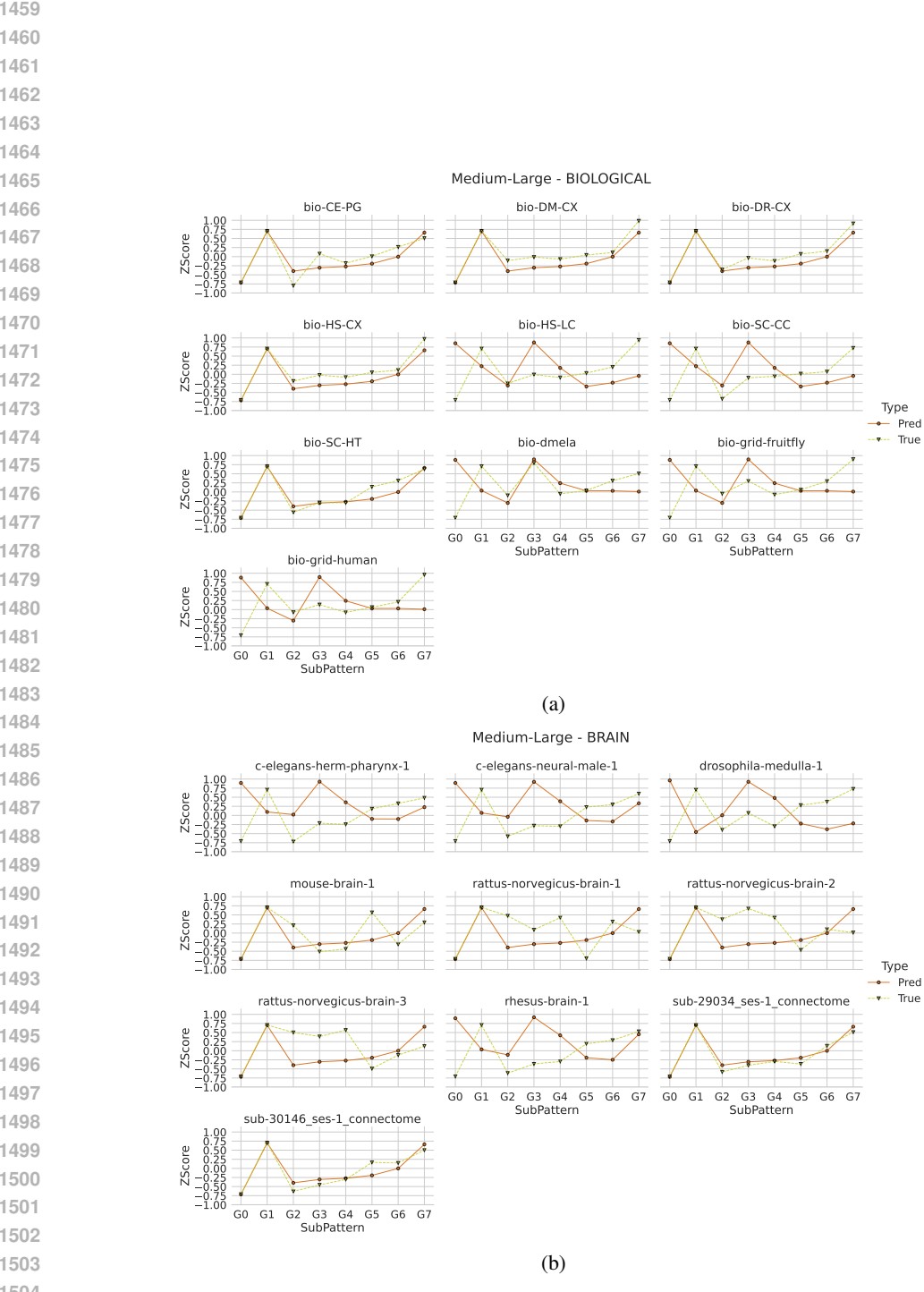

(a)

(b)

Figure 9: Continued on next page.

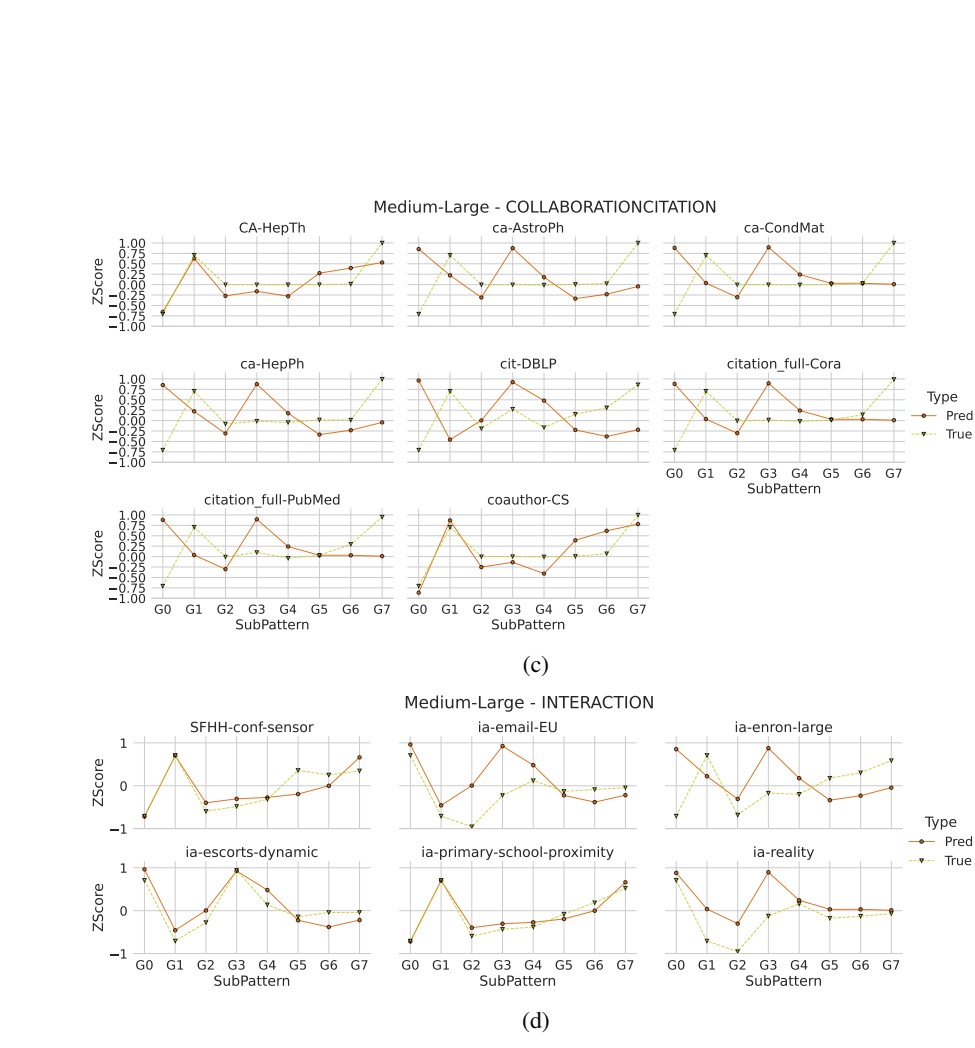

Figure 9: Predictions by GIN trained on the non-deterministic segment. Orange lines with circles are predictions and dark-yellow with triangles true values.

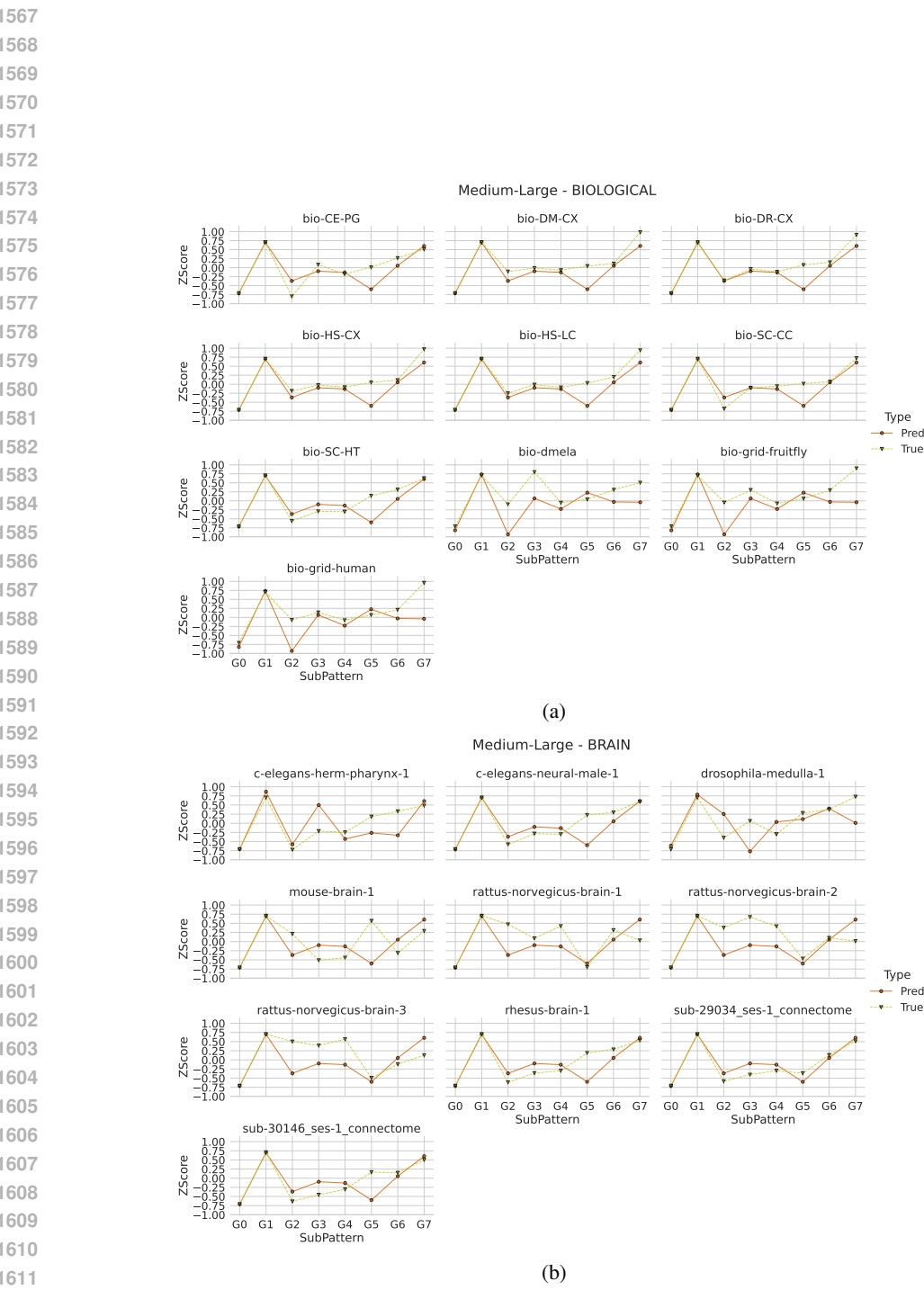

(a)

(b)

Figure 10: Continued on next page.

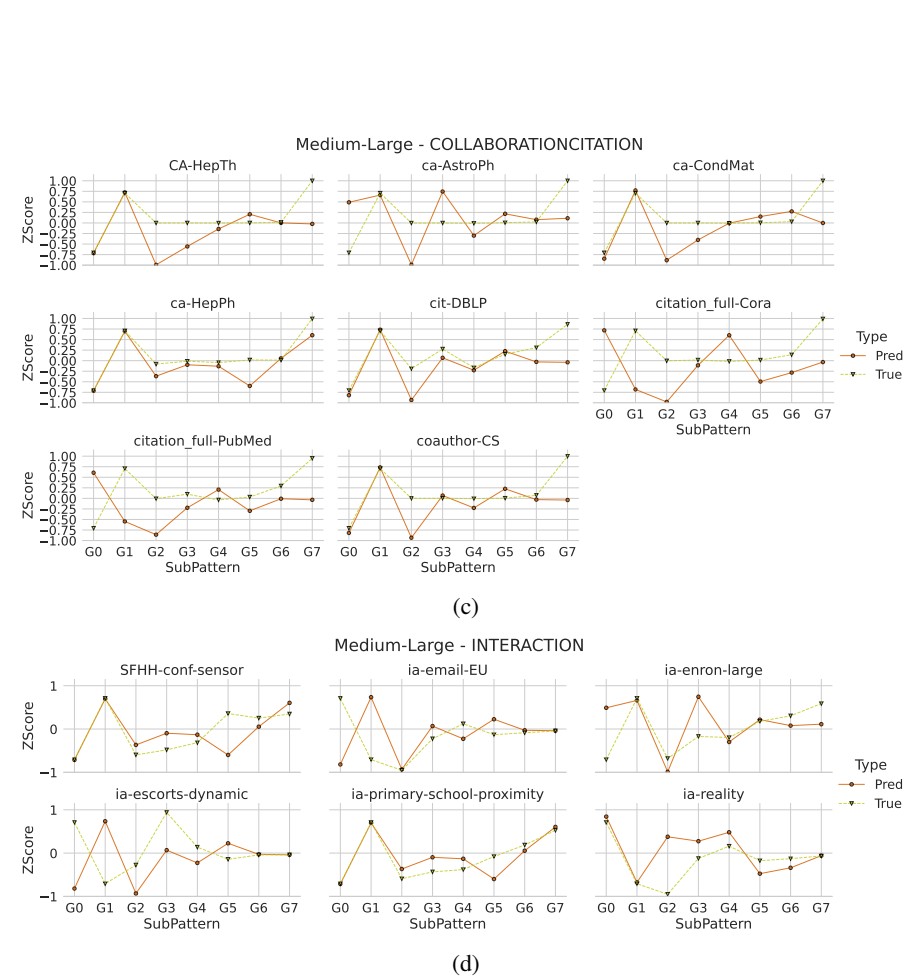

Figure 10: Predictions by GIN trained on the deterministic segment. Orange lines with circles are predictions and dark-yellow with triangles true values.

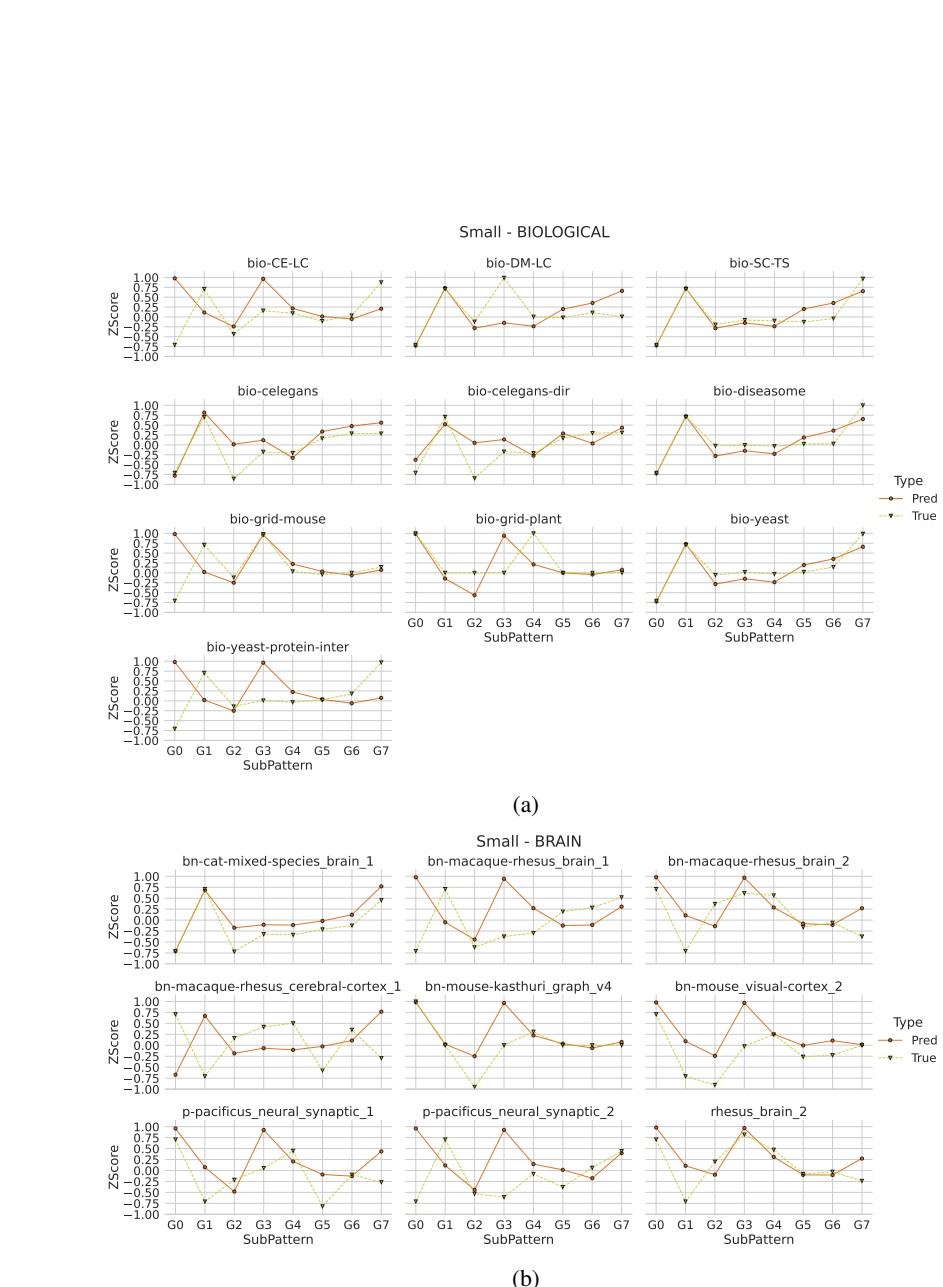

Figure 11: Continued on next page.

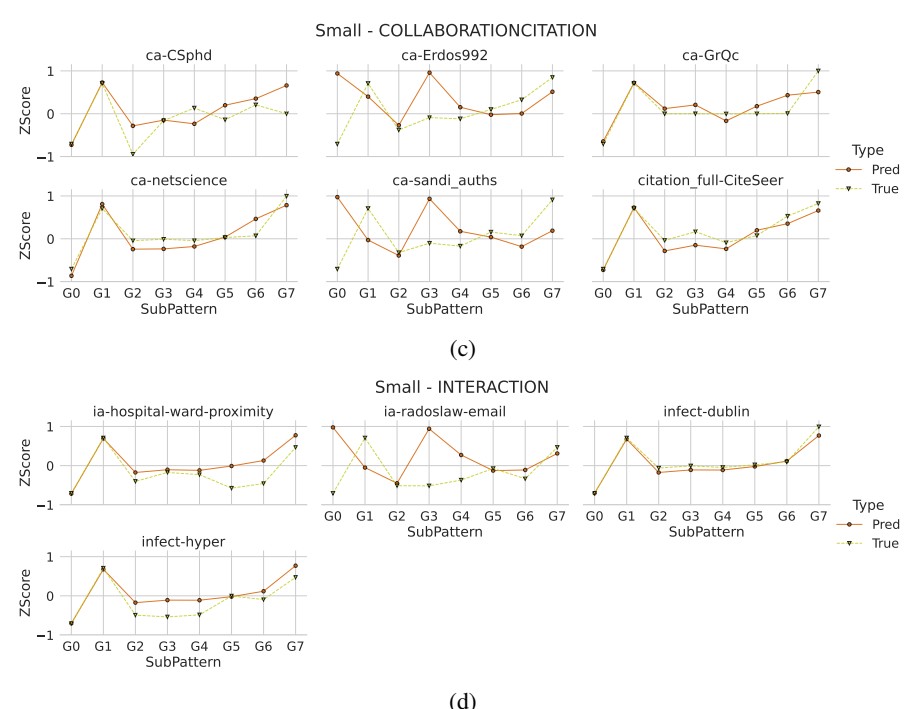

Figure 11: Predictions by GIN trained on the deterministic segment. Orange lines with circles are predictions and dark-yellow with triangles true values.

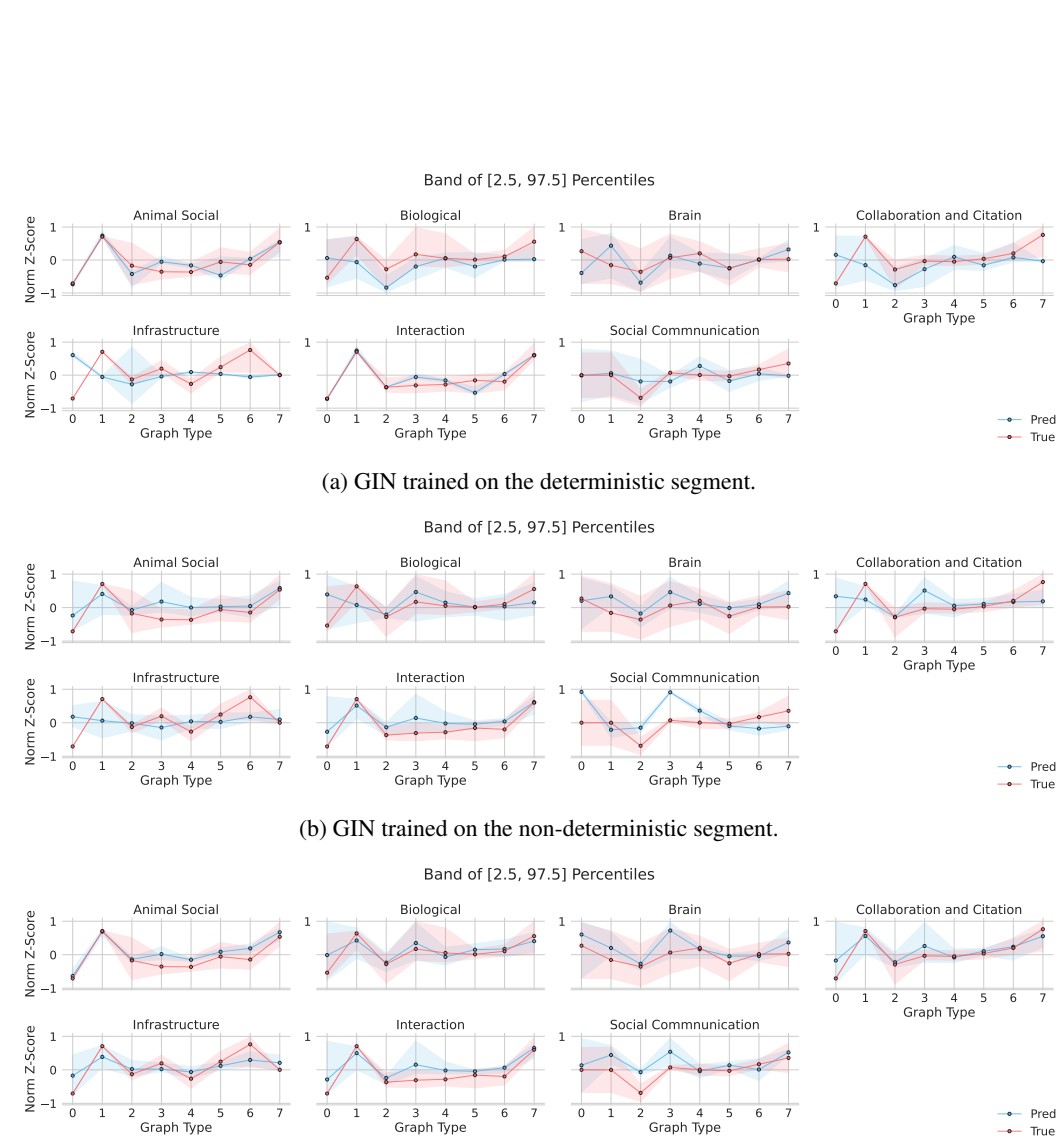

(a) GIN trained on the deterministic segment.

(b) GIN trained on the non-deterministic segment.

(c) SAGE trained on the non-deterministic segment.

Figure 12: Predictions for each model in for the small real-world dataset.

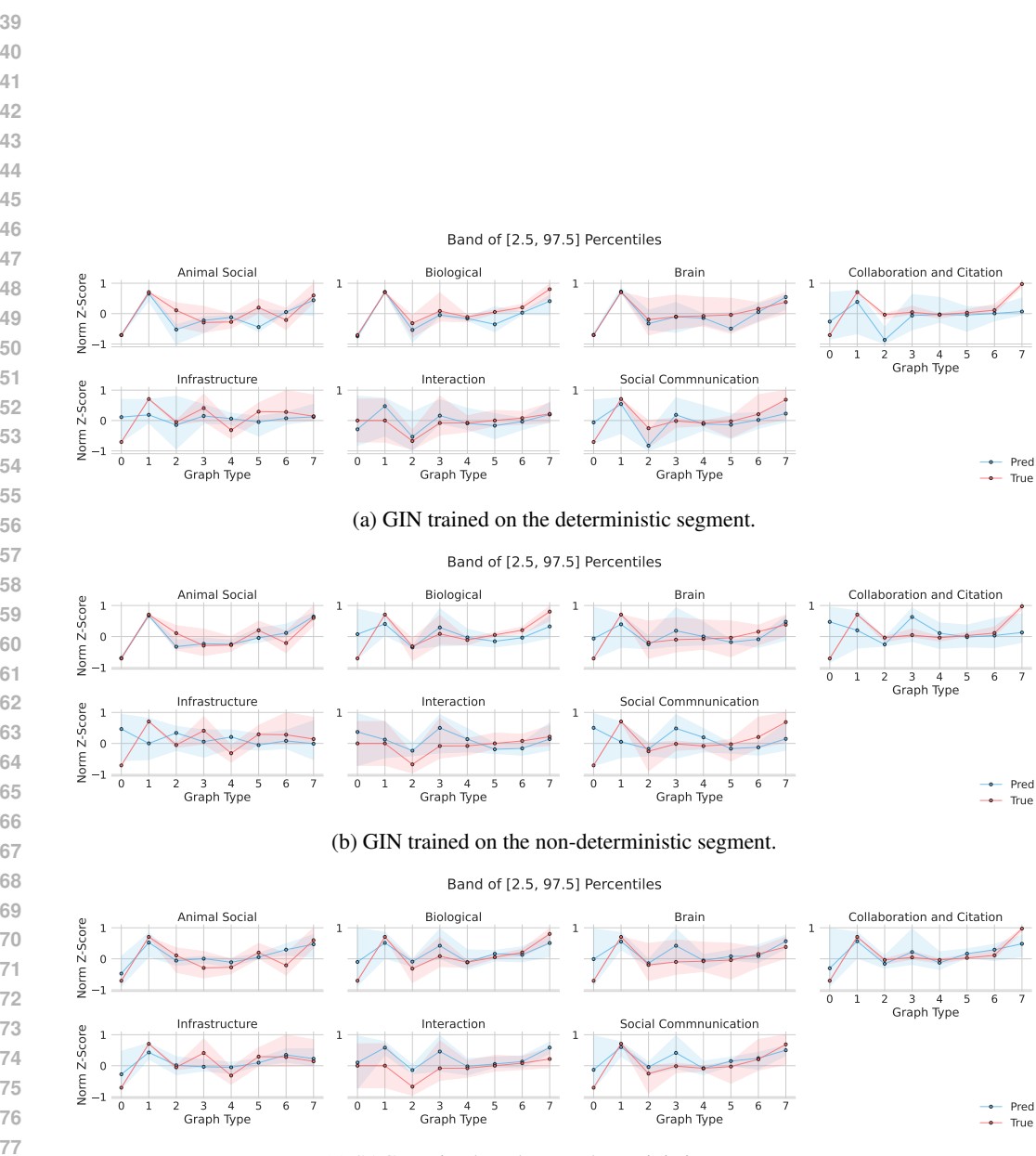

(a) GIN trained on the deterministic segment.

(b) GIN trained on the non-deterministic segment.

(c) SAGE trained on the non-deterministic segment.

Figure 13: Predictions for each model in for the medium-large real-world dataset.

