# OpenReview forum: "Predicting Network Motif Fingerprints with Graph Neural Networks"
_ICLR.cc/2025/Conference — ICLR 2025 Conference Withdrawn Submission_

### Official Review · Reviewer_QMx2 · 2024-10-30

**Soundness:** 2
**Presentation:** 1
**Contribution:** 2
**Rating:** 3
**Confidence:** 3

**Summary:**

The paper proposes a framework to extract network motifs, i.e., subgraph patterns that have some kind of significance, with graph neural networks. They formulate the task as a multitarget regression problem, where the targets are the significance scores of the (fixed) pattern graphs. The authors also provide a dataset of synthetic graphs that contain specific types of motifs/patterns.
In a nutshell, the topic is very interesting but I believe that the contributions of this paper are too limited for ICLR.

**Strengths:**

- the motif mining problem is a very relevant task, that has multiple applications in computational chemistry as well as social network analysis. Moreover, the approach proposed by the authors, albeit not very novel, is reasonable and seemingly effective.
- The code to generate the large and diverse dataset of synthetic graphs created by the authors could be useful for subsequent articles on motif discovery.

**Weaknesses:**

- The idea is not particularly novel, as the model seems to be (although the description is not very clear) just a GNN trained to regress a vector of significance values for a set of patterns of interest.
- The model can only obtain the significance of a fixed and pre-determined set of patterns, which limits the applicability of the method. In particular, the authors focus on unlabeled graphs of size 3 and 4. In practice, this is not ideal, as node labels carry a lot of information (e.g., in molecules, unlabeled patterns have little meaning, while labeled patterns can identify functional groups). Since the number of labeled patterns grows very fast, one could train the model only for a small fraction of them, and most of the subgraph pattern queries would not be able to be answered by the proposed model. If the model was able to learn the interesting motifs directly, the model could be much more useful.
- The authors give an unclear discussion in Section 4.2 of the relation between the ability of GNNs to count subgraphs and their task, but the explanation of why 1-WL MPNNs are sufficient for the motif mining task is not satisfactory and vague (“MPNNs remain highly valuable and find numerous practical applications in real-world scenarios”). It would be helpful if the authors provided proof of their statements or at least some experimental evidence.
- The experimental results are not only unsatisfactory (see point below), but they are also very hard to understand (e.g., I can’t understand the results in Table 1). The definition of “correct” and “incorrect” is unclear, and should be clarified and formalized better.
- The model has performance on real-world data that is close to the one of a model guessing at random and is much worse than the results on the synthetic data. This is a severe limitation to the applicability of the model to any real word task.
- Overall, the paper feels unpolished. Many statements are too vague or unclear. See some of my questions below.

#### Minor remarks:
- line 491: you have an extra comma.
- I appreciate that the authors provide the code to generate the synthetic datasets, but it is unclear whether a precomputed version is available for download.

**Questions:**

- line 196: “Thus, we start with a vector of Z-scores”. You mean that the vector is your ground-truth label? This is unclear
- the paragraph on “Correct predictions, line 340” is very unclear to me. Can you please elaborate on your metrics?
- line 458: “due to failing to approximate an injective function or because the bound does not work well in practical scenarios.” Which bound are you referring to? This needs further elaboration.
- which null model are you using to generate the ground truth significances? I can’t find this in the paper.
- what is the “margin of error” that you mention?
- Do you have any insights on how to obtain better performance on real world data?

---

> ### Author Response · Authors · 2024-11-24
> **Response to reviewer QMx2**
>
> We want to thank the reviewer for their time, valuable comments and outstanding effort. The comments provided are very constructive.
>
> 1. Regarding the first point in the weaknesses, we acknowledge that the model is not novel and apologise if our writing implies otherwise. As pointed out in line 323, the used model is essentially the one from [1] definition A.1. from Appendix A. Our focus is on the method/formulation and not on the model.
> ---
> 2. Regarding point 3., we believe will add references for our claims in a further iteration of the manuscript.
> ---
> 3. Regarding point 5., we acknowledge that the performance on real-world data is not as desired. However, while the MSE is close to that of a random model, this does not imply that the predictions are as uninformative as those from a random model. This might feel counterintuitive and we tried to describe this fact through lines 390-401 and then by answering questions (1-3) from lines 402-404. In summary, "(...) while a trained model predicts meaningful patterns ineffectively, a random one only predicts highly persistent patterns, leading to a similar average scores." (line 514). Concretely, the predicted patterns for real-world networks typically follow that of the mean pattern of the synthetic generator that most resembles the real-world network (line 495).
> ---
> 4. The precomputed dataset is available for download as indicated in the readme. Since the readme might be overly detailed, locating the links quickly might be challenging. We will make adjustments so that they are easily seen.
> ---
> 5. Regarding question 1, some questions regarding what the target might permeate by reading line 196. This was not our intention, the expression is meant to indicate that the starting point for the subsequent reasoning starts with a vector of Z-scores. Lines 233-236 make it clear what the target is.
> ---
> 6. Regarding question 2, the metrics for a correct prediction are formed according to two principles: (1) have a systematic method that does not depend on the research field (and type of network used); (2) one prediction, to be considered correct, must always predict over-representation/under-representation correctly for all 8 graphs (because this is a key concept). To accomplish point (1), since we lack a "magic number", for a model predicting $\mathbb{\hat{S}} = \{s_1, \dots s_n \}$, we opt to select correct predictions such that: $\text{correct} = \\{s | s_i \leq m,  i \in |\Omega|,  \forall s \in \mathbb{\hat{S}} \\}$ with $m = MSE(\hat{\mathbb{S}}, \mathbb{S})$ and $\Omega$ the set of graphs used. This criterion allows us to select only predictions with a loss no greater than the global loss for graphs in $\Omega$. Hence, the margin of error is dependent on the general capability of the model, more capable models will have more tight margins. After reading your comment, we reflected on this margin and acknowledged the shortcomings it has in model comparison. We will reformulate this in a further iteration of the paper.
> ---
> 7. Regarding question 3, our wording is unclear, we will rectify it. We refer here to the bound established in [2,3], which suggests that for random graphs of size $n$, as $n \rightarrow \infty$, CL algorithms should be able to distinguish them.
> ---
> 8. Regarding question 4, the NULL model used is illustrated in line 138. We acknowledge that this may not have been sufficiently clear.
> ---
> 9. Regarding question 5, the margin of error would be $m$ from point 6.
> ---
> 10. Regarding question 6, we believe that future work should focus on increasing the expressivity of the backbone model. Upon a closer inspection to the data, even though the synthetic dataset is larger and more diverse, it is also much more dense. Hence, the big discrepancy is mainly due to the synthetic data having many different networks that end up having very similar SPs. Hence, the performance appears better because the mean profile is strong. In the real-world dataset, the SPs are more sparse, leading to larger errors. Furthermore, for the problem at hand, we believe that as long as the model $M$ used has dimensionality $<4$, our formulation of directly estimating Z-scores and using multitarget regression should further boost the results of $M$.
>
> \
> \
> References:\
> [1] Zhengdao Chen, Lei Chen, Soledad Villar, and Joan Bruna. Can graph neural networks count substructures? \
> [2] László Babai and Ludik Kucera. Canonical Labelling of Graphs in Linear Average Time. Annual Symposium on Foundations of Computer Science - Proceedings, (2):39–46, 1979. ISSN02725428. doi: 10.1109/sfcs.1979.8. \
> [3] László Babai, Paul Erd˝os, and Stanley M Selkow. Random Graph Isomorphism. SIAM Journal on Computing, 9(3):628–635, aug 1980. ISSN 0097-5397. doi: 10.1137/0209047

---

> > ### Comment · Reviewer_QMx2 · 2024-11-25
> >
> > Dear authors,
> >
> > Thank you for your response, after reading your clarifications some points of the paper are now easier to understand. Including these clarifications in the next iteration of the paper will surely make it easier to follow.
> >
> > With regards to point 10, I also believe that both addressing the expressivity of the model and making the distribution of the synthetic data match more closely the real world data could lead to improved results. You might also try to pre-train on this synthetic data and then finetune the model on real world data.
> > Best of luck for your next submission.

---

### Official Review · Reviewer_uysJ · 2024-10-31

**Soundness:** 2
**Presentation:** 2
**Contribution:** 2
**Rating:** 3
**Confidence:** 4

**Summary:**

This paper tackles the challenge of predicting network motifs using Graph Neural Networks (GNNs). Given the known expressiveness limitations of GNNs, the authors reframe motif estimation as a multitarget regression task, where curated regression targets are assigned to each input graph. This method is validated on a large, diverse synthetic dataset. The study reveals that 1-WL-limited models trained on synthetic data struggle to generalize effectively to more complex, real-world networks.

**Strengths:**

- This paper tackles an important and challenging problem of using GNNs for network motif estimation.

**Weaknesses:**

- The proposed workaround—framing motif prediction as a multitarget regression task instead of directly predicting network motifs—relies heavily on heuristics and lacks sufficient motivation. Neither theoretical justification nor empirical evidence is provided to explain why altering the task would enable GNNs to perform better. For example, the statement in line 275, "We might have made the problem easier (or harder) than substructure counting," adds to the ambiguity surrounding the authors’ approach.

- As a heuristic-based study, the experiments are restricted to synthetic data, limiting the findings’ relevance to practical applications in complex real-world networks.

- While acknowledging the inherent limitations of GNNs, the authors overlook recent approaches leveraging large language models for classical graph problems like subgraph counting (e.g., [1]). Such methods, despite their own limitations, have shown performance improvements over GNNs but are neither cited nor compared here.

- Limited backbone choices: Only SAGE and GIN are used as backbones, while more expressive GNN models are not included (see, e.g., [2–4]).

[1] Understanding Transformer Reasoning Capabilities via Graph Algorithms.

[2] From stars to subgraphs: Uplifting any GNN with local structure awareness.

[3] Building powerful and equivariant graph neural networks with structural message-passing.

[4] Understanding and extending subgraph gnns by rethinking their symmetries.

**Questions:**

Aforementioned in weaknesses.

---

> ### Author Response · Authors · 2024-11-24
>
> We deeply appreciate the feedback and the response received from the reviewer.
>
> 1. Regarding point 3, rather than following all steps outlined in section 3.2 for motif analysis, we propose merging steps 2 and 3 and "(...) design a method for motif finding, leveraging a novel formulation that hinges primarily on reworking the target task to something else other than direct substructure counting." (line 49). Approaches using language models were not included in our comparison, as they are outside our focus on motif discovery methods. However, we acknowledge that language models as a medium for subgraph estimation could be inserted into section 3.2 (as for [2] specifically, at the time of writing, we were not aware of its existence).
> ---
> 2. Regarding point 4, we did test 2 more backbones GCN and GAT (line 1176 - still 1-WL-limited). Furthermore, following the reasoning from the previous point, it was not our main objective to compare our formulation across various backbones. Since our formulation is designed only with the constraint of having dimensionality $<4$, due to the size of the graphs selected, we expect any model following the proposed constraint to end up benefiting from the formulation. Nonetheless, we acknowledge that this claim should be supported by additional experiments on other backbones.
>
> \
> \
> References:\
> [1] Zhengdao Chen, Lei Chen, Soledad Villar, and Joan Bruna. Can graph neural networks count substructures? Advances in Neural Information Processing Systems, 2020-Decem(NeurIPS), feb 2020. ISSN 10495258\
> [2] Understanding Transformer Reasoning Capabilities via Graph Algorithms.

---

### Official Review · Reviewer_DuKG · 2024-11-02

**Soundness:** 1
**Presentation:** 1
**Contribution:** 1
**Rating:** 1
**Confidence:** 5

**Summary:**

The paper proposes a method that frames motif estimation as a multitarget regression problem, which enhances interpretability and scalability for large graphs. The findings indicate that while models trained on synthetic data perform reasonably well, they struggle with real-world networks, highlighting a gap in generalization.

**Strengths:**

1. The paper presents a unique formulation of motif estimation as a multitarget regression problem.
2. The paper sets a benchmark for future research in the field.

**Weaknesses:**

1. The introduction section requires significant improvement, as it contains poorly defined terms and vague sentences. For instance:
In line 40, the phrase “what is a good network model for protein-protein interactions” lacks clarity. The authors may have intended to ask which architecture is preferred for this purpose. Additionally, the term "control networks" in line 42 is not clearly defined.
The phrase “very hard computational task” in line 43 is not professionally articulated. The term "very" is ambiguous; it would be more precise to state that the task is NP-hard.
The statement in lines 46-48 lacks supporting references, making the claim questionable. Additionally, phrases such as “typically do not give very interpretable results” are vague. What exactly is meant by “very interpretable”? This concept requires clearer definition and explanation.
Lines 49-52, which aim to outline the main contributions of the paper, are overly vague. The term “interpretable scores” needs further clarification. In the context of motif discovery, what does “score” refer to? Furthermore, the phrase “ensuring the possibility of further insight into the conclusions obtained” is unclear; which conclusions are being referred to?
Line 43 states, “Additionally, our method is robust and versatile, capable of operating effectively on graphs of any size.” The term "robust" is ambiguous in this context, and since many graph neural networks (GNNs) can operate on graphs of varying sizes, this statement does not clearly highlight the uniqueness of the authors' approach.
The sentence in line 53, “Knowing how difficult it is to obtain a high volume of real-world graph datasets that have both high quality and variety,” lacks supporting references, which undermines the claim.
Finally, lines 62-63 are poorly constructed: “We make available a large diverse synthetic dataset in terms of graph topology and motif significance-profile generated with 23 synthetic generators.” This statement could benefit from clearer phrasing and more specificity regarding what is meant by “motif significance-profile.”
More vague and poorly written lines: 192-193.
Overall, the introduction requires substantial revision for clarity, precision, and supporting evidence.


2. The motivation for the proposed method is unclear, primarily due to the lack of references supporting the claims made in the introduction (lines 46-48). Additionally, Section 3.2 outlines numerous methods for detecting substructures and motifs but fails to address their limitations or specify how the proposed method intends to resolve these issues.

3. The structure of the paper could be enhanced for better clarity and coherence. For example, dedicating an entire section (Section 2) to preliminaries that consists of only four lines seems excessive, especially since these preliminaries are only referenced in Section 4. This placement raises questions about its necessity. Additionally, the title of Section 4, “INITIAL PROBLEM DESCRIPTION,” is misleading, as the majority of the section focuses on the proposed method rather than outlining the problem. Furthermore, lines 180-188 appear to contain preliminary information; starting a section with such lines without appropriate context is unprofessional and detracts from the overall readability of the paper.

4. Ultimately, the method is limited to small graphs of sizes 3 and 4, which restricts its overall contribution. Additionally, there is a lack of explanation regarding why existing approaches fail to detect such graphs, as previously mentioned in comment 2.

5. While the authors assert that interpretability is one of the main contributions of the paper in the abstract and introduction, the topic is not addressed further, nor is there any demonstration of how the method achieves interpretability. In the context of explainable artificial intelligence (XAI), which emphasizes transparent, white-box models, it is unclear how the proposed approach qualifies as interpretable. Additionally, the paper fails to establish how their method offers enhanced interpretability compared to existing approaches.

6.The paper does not provide any comparisons to existing approaches, which is a significant oversight. Without such comparisons, it is difficult to assess the effectiveness and advantages of the proposed method in relation to established techniques. Including a comparative analysis would enhance the paper's contributions and provide clearer context for the claims made about the method's performance and interpretability

**Questions:**

1. What is the motivation behind the proposed method? Specifically, what problems associated with existing approaches does it address, particularly in relation to the methods discussed in Section 3.2?
2.The method focuses on small graphs of sizes 3 and 4. What are the implications of this limitation for the broader applicability of your approach
3.Can you provide more insight into the limitations of existing methods for detecting small graphs? What specific challenges do they face that your method aims to address?
4.How does your method enhance interpretability compared to existing approaches? Can you provide specific examples or references to studies that illustrate the limitations of current methods in this regard?
5.I kindly request that you include a comparative analysis of your proposed method against existing approaches.

---

> ### Author Response · Authors · 2024-11-24
> **Response to reviewer DuKG pt1**
>
> We appreciate the feedback and the response received from the reviewer.\
>
> 1.
>  - In line 40, we will look into changing the expression used. What was meant by the expression used is that graphlet degree distribution has been used to understand what network model e.g. scale-free, geometric is a good fit to represent protein-protein interactions.
>  - Regarding line 43 we will reformulate it so that it is more clear that the problem is at least NP-complete as it described in line 44 (we would rather not address it as NP-Hard since it can create confusion if not explicit that it is also NP-complete).
>  - Regarding line 52 (probably a typo in the review to line 43) while it is true that many GNNs can operate in graphs of varying sizes, many cannot. Hence, why we state that our method can work in graphs of multiple sizes.
> ---
> 2. In section 3.2 we outline multiple methods for counting substructures, not to estimate motifs. We acknowledge that these are distinct concepts and will ensure this distinction is emphasised. The methods highlighted in section 3.2 can be used as a part of the process for estimating motifs by using them to count subgraphs as stated in section 3.4. We address the issues with methods that estimate motifs directly in section 3.4.
> ---
> 3. We agree with most of the statements regarding section 4 and will revise them. However, we placed lines 180-188 where they are to support the progression of ideas without disrupting the flow of earlier sections. Nonetheless, we are open to exploring the point of view of the reviewer regarding the usage of proper context and relocating this information. However, we ask if the reviewer could clarify their intended meaning of 'appropriate context.'.
> ---
> 4. We apologise for a possible misunderstanding that could have been caused by some ambiguities in the text but we are not comparing our method directly with the methods highlighted in section 3.2. These methods, and GNNs in general, can be used for a panoply of tasks one of them that is commonly seen in the literature (section 3.2 - GNN Methods.) is subgraph counting. Subgraph counting is one of the steps used for motif estimation (section 3.2). We propose an alternative to the steps in section 3.2, and instead  "(...) design a method for motif finding, leveraging a novel formulation that hinges primarily on reworking the target task to something else other than direct substructure counting." (line 49). We then proceed to elaborate on how this can be achieved (section 4) and analyse if there are any advantages to such reformulation (section 7). We conclude that despite the advantage existing, it is not enough for a 1-WL-limited model trained on synthetic data since it struggles to generalise effectively to more complex, real-world networks. The intent of the analysis of these results was to delve into the underlying reasons behind these outcomes and to offer readers a structured understanding of why such generalisation issues occur and what they represent.
> ---
> 5. We hope to clarify that in this work, ‘interpretability’ specifically refers to the interpretability of the model's output, not to model interpretability as in explainable AI. Typically, in the relevant literature, [4,5], the motif problem is formulated so that the outputs are a set $S = {S_1, \dots, S_n}$ of graphs that the model considers as a motif. However, following traditional literature regarding motif analysis [1,2,3], these results lack interpretability since there is no clear indication of how a graph $S_x$ compares to $S_y$. That is, even though these two were considered motifs, the methods offer no insight into questions like Is $S_x$ more overrepresented than $S_y$? We aim to retain interpretability by outputting a Z-score, providing insight into overrepresentation among motifs, which is relevant for motif analysis.
> ---
> 6. This point is somewhat tricky. We do not provide a direct comparison with other methods because, as far as we are aware, there are no other methods. As described in point 5., the approach to motif discovery using GNNs is commonly focused on obtaining a set $S = {S_1, \dots, S_n}$ of graphs that the model considers as motifs. Hence, comparing with these methods is far from trivial (if even possible to have a reliable method for such comparison). Since we focused on reformulating the base problem, the most we could do was to compare it with the previous formulation (section 7.5). Furthermore, we did not perform more experiments with the models from section 3.2 because they are only relevant as a backbone, meaning that, for the case of subgraphs of size $\leq4$, any advantage obtained by an MPNN should also be obtained by any other GNN model as long as it has dimensionality $<4$ [6]. Nonetheless, we acknowledge that this claim should be supported by additional experiments on other backbones.

---

> ### Author Response · Authors · 2024-11-24
> **Response to reviewer DuKG pt2**
>
> Questions:
> 1. We believe that our answers to points 4-6 address the reviewer’s concerns in Questions 1,3,4,5.
> 2. Line 212 addresses why we picked small graphs of sizes 3 and 4. Our approach concludes that there is a benefit in combining the steps for motif discovery in a single task for sizes 3 and 4, meaning methods with dimensionality $<4$ should benefit from this approach. However, for a general graph size on a general output vector, since a part of the benefit stems from selecting graphs that are related to each other, we cannot predict how the results would be.
>
> \
> \
> References:\
> [1] Ron Milo et al., Superfamilies of Evolved and Designed Networks. Science 303,1538-1542(2004).DOI:10.1126/science.1089167\
> [2] Shen-Orr, S., Milo, R., Mangan, S. et al. Network motifs in the transcriptional regulation network of Escherichia coli. Nat Genet 31, 64–68 (2002). \
> [3] R. Milo, S. Shen-Orr, S. Itzkovitz, N. Kashtan, D. Chklovskii, and U. Alon. Network Motifs: Simple Building Blocks of Complex Networks. Science, 9781400841(October):217–220, 2004a. \
> [4] Carlos Oliver, Dexiong Chen, Vincent Mallet, Pericles Philippopoulos, and Karsten Borgwardt. Approximate Network Motif Mining Via Graph Learning. 2022.\
> [5] Shichang Zhang, Ziniu Hu, Arjun Subramonian, and Yizhou Sun. Motif-Driven Contrastive Learning of Graph Representations. Proceedings of the AAAI Conference on Artificial Intelligence, 35 (18):15980–15981, dec 2020. ISSN 2374-3468. doi: 10.1609/aaai.v35i18.17986 \
> [6] Matthias Lanzinger and Pablo Barceló. On the power of the weisfeiler-leman test for graph motif parameters, 2023\

---

> > ### Comment · Reviewer_DuKG · 2024-11-26
> >
> > I thank the authors for their response. Nonetheless, I believe the promised changes are major revisions, and that this paper will benefit from a re-submission after dressing the points raised by the reviewers here, and restructuring the paper.
> > I still find it hard to see the benefit of the proposed view without a direct comparison to other approaches. If as the authors claim no such method exists to compare it to, it is recommended therefore to better show how the proposed idea solves something that is really critical for the community or a diverse set of problems.

---

### Official Review · Reviewer_LUCZ · 2024-11-04

**Soundness:** 2
**Presentation:** 3
**Contribution:** 2
**Rating:** 3
**Confidence:** 4

**Summary:**

The paper proposes a GNN model for estimating the frequency of a set of subgraphs (called motifs) in a given graph.
The described estimator predicts the standardized occurrence counts (Z-counts) $z_i$ for subgraph H_i in a target graph $\mathcal{G}$ relative to a set of reference graphs $\mathcal{G}_\mathrm{NULL}$, i.e., the model is trained to predict how many standard deviations the occurrence count of H_i in $\mathcal{G}$ is above or below that in G_NULL.

The proposed approach normalizes the Z-counts to the interval $[-1, 1]$. Additionally, the Z-count vector $z$ is normalized via the Euclidean norm to obtain a so-called *significance profile* vector $s$.
The model is trained by minimizing the MSE between the true and the predicted significance profiles of synthetically generated training graphs.

The model is evaluated by comparing the significance profile MSEs on unseen synthetic and real-world graphs. The proposed approach performs well on the synthetic validation and test data. However, on real-world graphs, the estimator performs significantly worse.

**Strengths:**

The proposed approach is well motivated and contextualized by giving a good overview of the strength and weaknesses of prior work in the field.

The idea of predicting Z-scores instead of absolute occurrences to circumvent potential generalization  issues with directly predicting absolute counts which might grow super-exponentially is convincing.

Additionally, the relations between absolute counts and significance profile estimation described in Section 4.2 raise interesting questions about the possibility of difficulty classes for different counting problems wrt. different NULL models.

The analysis of the reported empirical results is thorough with many details being provided in the supplement.

**Weaknesses:**

As described by the authors, the proposed approach does not appear to produce satisfactory results on real world data. However, since "negative" results can also be of interest, I will not focus on that here.

First, while theoretically interesting, I found it difficult to assess to which degree the proposed approach might be useful for practical purposes, mainly due to the following reasons:
1. The approach is only compared to itself. As described in Section 3, there has been prior work on substructure counting with GNNs. An empirical comparison to another counting model (e.g., [1]), would have been helpful.
2. The performance of the estimator is only evaluated wrt. the MSE of the predicted significance profiles. While interesting, it is unclear to me to which extent those significance profiles are useful in downstream tasks. Why was the quality of the predicted Z-scores or possibly the absolute occurrence counts not evaluated?
3. Related to my previous point, it would have been interesting to highlight potential applications for the proposed approach in the experiments, e.g., via a case study showcasing how the estimator might be used in practice.

Second, I found the motivation for using significance profiles at all to be lacking.
In Section 4.1 the authors write that this "imposes a mathematical interconnectivity between the Z-scores"; no explanation as to why such a constraint is desirable is given.
For induced subgraphs one can argue that an increase of one subgraph of a certain size might lead to a decrease in occurrence of other subgraphs of that size, assuming that the set of motifs contains **all** subgraphs of a given size.
However, since this is a constraint on the sum of the Z-scores, L1 normalization instead of the used Euclidean norm would have been appropriate.
Can you elaborate on why you chose to use L2-normalized significance profiles and discuss the advantages (and potential disadvantages) of this choice?

Third, in Section 7.2 conjecture that the GNN models used in the estimator might not be expressive enough to properly learn occurrence counts in the evaluated setting.
Given this observation, it is unclear why the authors did not also evaluate more expressive GNN models.
In Section 4.2, they mention that more expressive models have high computational complexity; however, there are a number of higher-order GNN architectures with different performance characteristics [2,3,4], e.g., 2-FWL approximations with a tunable complexity parameter [3].
Even if all currently proposed higher-order approaches were infeasible for the intended setting, a small scale evaluation of more expressive models would have been interesting.
Could you run such an experiment, e.g., using one of the mentioned higher-order GNN approaches?

Last, I have a few minor comments regarding the presentation:
- Figure 1 is difficult to parse by itself; more specifically, repeating the x-labels on all six subplots and adding a meaningful y-axis label would be helpful.
- In Table 1, the distinction between SAGE and GIN using underline and italics is unnecessarily hard to parse; adding an explicit heading to the already existing columns for both would be better.
- On page 2, line 54, I found the choice of the word "myriad" to be vague and misleading, when the number of used generators is "just" 23.
- On page 6, line 304: Grammar. "being those" should read "those being".

---
[1] Chen, Z., Chen, L., Villar, S., Bruna, J.: Can graph neural networks count substructures? In: Proceedings of the 34th International Conference on Neural Information Processing Systems. pp. 10383–10395. Curran Associates Inc., Red Hook, NY, USA (2020).

[2] Maron, H. et al.: Provably Powerful Graph Networks. arXiv. (2019).

[3] Damke, C. et al.: A Novel Higher-order Weisfeiler-Lehman Graph Convolution. In: Proceedings of The 12th Asian Conference on Machine Learning. pp. 49–64 PMLR (2020).

[4] Tahmasebi, B. et al.: Counting Substructures with Higher-Order Graph Neural Networks: Possibility and Impossibility Results, [http://arxiv.org/abs/2012.03174](http://arxiv.org/abs/2012.03174), (2021). [https://doi.org/10.48550/arXiv.2012.03174](https://doi.org/10.48550/arXiv.2012.03174).

**Questions:**

A few questions were already raised in the weaknesses, namely:
- Why were no other occurrence estimations approaches used for comparison in the evaluation?
- Which downstream applications do you envision for your proposed approach?
- Why the focus on significance profiles instead of Z-scores?
- Why are the significance profiles normalized using L2 instead of L1?

Additionally, out of curiosity, I would like to ask why the set of significance values specified in Section 5 (line 305) is as described. The explanation for why $\frac{1}{\sqrt{2}}$ appears is understandable, but why are 0 and 1 also part of the set and why is -1 not in it?

---

> ### Author Response · Authors · 2024-11-24
> **Response to reviewer LUCZ**
>
> We want to deeply thank the reviewer for their time, valuable comments and outstanding effort. The comments provided are very constructive and the suggestions very relevant.
>
> 1. Regarding the first issue pointed out, since this was highlighted consistently by the reviewers, we may not have communicated our experiments clearly. We sincerely apologise for that. We did a comparison against a model directly counting subgraphs. Such comparison is summarised in section 7.5 and explained with greater detail in Appendix D.1. We did not use multiple models because our formulation is designed to benefit any model with dimensionality less than 4, a constraint delimited by the size of the selected graphs. Regardless, we acknowledge that this claim should be supported by additional experiments on other backbones.
> ---
> 2. As far as we are concerned, there is not a widely accepted method to assess the quality of predicted Z-scores for motifs. In the paper, we acknowledged the limitations of MSE. This was one of the main factors driving the analysis made in section 7.1 and the following sections. We did try to make another type of analysis by clustering predicted Z-scores, first mentioned in the paragraph from line 331 and the one from 390. However, following your comments, we believe this analysis could be refined for clarity and effectiveness.
> ---
> 3. Thank you for your suggestion, we will incorporate potential applications in a further iteration of the paper.
> ---
> 4. Regarding your comment on the expression "imposes a mathematical interconnectivity (...)", in line 231, this interconnectivity is desired because: "Should the selected graphs exhibit negligible relation, an attempt to predict the Z-score concurrently for all graphs may prove harmful to the performance of the model." (line 202). By making the scores for the graphs of the same size be **clearly** related and dependent on each other we impose a very strong constraint to the space of possible valid significance profiles. We do not deny the possibility of the existence of such a relation even if the scores are not normalised, but such dependence would be much more complicated. Furthermore, the normalisation of scores allows for the comparison of the motif significance profiles across networks of different sizes. Regarding the usage of L2, on top of being one of the standards for motif normalisation [1], we did not foresee any significant drawback when compared with L1 normalisation. In fact, from our tests, L2 seems to offer a better compromise between a good spread of points and a limited presence of extreme values.
> ---
> 5. As mentioned in point 1., we expect any model following the proposed constraint to end up benefiting from the formulation. Following your indications, we are currently trying to validate this claim. Initially, we did not include these experiments since other models we tried were too inefficient to be used with the hardware we had available. Furthermore, we opted to not include tests on a smaller scale since we, at the time, believed it would defeat the purpose of the tests with the complete dataset.
> ---
> 6. First question please refer to points 1 and 5.
> ---
> 7. We expect our work to show a technique that can enhance the capability of GNN models for motif estimation. When using GNN models, it is inefficient and even harmful to go through all the steps used in traditional motif estimation. Furthermore, we aim to demonstrate empirically that MPNNs are unreliable for general motif estimation. The general quality of the predictions is only good enough to approximate the graph generator model. Hence, applications where a quick network model estimation is necessary could still benefit from the capabilities of MPNNs.
> ---
> 8. Given that networks of different sizes can have subgraph counts that result in a high disparity in the magnitude of Z-scores, the normalisation emphasises the relative rather than absolute importance of subgraphs. This allows a better comparison between networks of different sizes.
> ---
> 9. Fourth question please refer to point 4.
> ---
> 10. The appearance of 0 and 1 is "(...) an artefact from the G-Trie model, primarily affecting the duplication-divergence model (...)" line 307. These values result from an unreported artefact or 'bug' in the G-Trie software. It happens when the model cannot construct randomised networks for that degree sequence within a certain number of tries. Even though the number of affected graphs is **very** small we decided to include this mention since the dataset made available still includes these instances.
>
> \
> \
> References:\
> [1] Ron Milo et al., Superfamilies of Evolved and Designed Networks. Science 303,1538-1542(2004).DOI:10.1126/science.1089167

---

> > ### Comment · Reviewer_LUCZ · 2024-11-26
> >
> > Thank you for your detailed response! Due to the withdrawal, I will not go over it point by point, but overall, my questions were mostly answered.
> >
> > Out of interest, I only have one follow-up question regarding point 4:
> > While it is very plausible that the L2 normalized profiles work well in practice, my question was primarily about as to *why* this might be the case. While possibly out of scope for your work, I believe it would be interesting to understand why and under which assumptions normalization works. Could you perhaps share an intuition or simple example that sheds some light on how motif occurrences might be related wrt. to Euclidean distances?
> >
> > I am looking forward to a future iteration of the paper that incorporates the suggestions made.

---

> > > ### Author Response · Authors · 2024-11-27
> > >
> > > Before normalising, let $\boldsymbol{x} \in \mathbb{R}^n$ be the vector of occurrences of $n$ graphs picked as candidate for motifs analysis. This vector alone carries no information regarding motifs. In order to perform motif analysis, it is needed to have other networks for comparison. To make this comparison it is necessary to select a characteristic of interest since networks can be similar at different multiple levels e.g. degree distribution, diameter, clustering coefficient etc. Given a fixed characteristic of interest, since it is still possible for other characteristics to change, multiple networks are generated. We assume the networks generated from the null model based on the chosen characteristic and the original network come from the same population. Hence, we can evaluate how $\boldsymbol{x}$ is positioned in the distribution, giving us a notion of over or under representation. The whole process, standardization and normalisation works if the previous assumption holds, that is, the initial network of interest and the control networks generated come from the same distribution wrt. to the characteristic of interest. Let $\boldsymbol{z}\prime$ be standardized form of $\boldsymbol{x}$. In order to ensure invariance to the magnitude of the networks, we normalise $\boldsymbol{z}\prime$ resulting in $\boldsymbol{z}$. We could think of $\boldsymbol{z}$ as features that embed the original network in a $n$ dimensional hypersphere. The distance between two networks $\boldsymbol{z}_1$ and $\boldsymbol{z}_2$ can quantify the similarity between them in terms of how structurally similar (specifically wrt. the original $n$ graphs) they are in the embedding space. Comparing the distance across specific dimensions allows a fine grained comparison between specific graphs - a small distance between $\boldsymbol{z}_1$ and $\boldsymbol{z}_2$ in the first dimension indicates that they are similar in how the number of 3-paths. For example, if both are overrepresented and with a similar score, these networks might have a close relationship on how scarce triadic closure is.
> > > Regarding the task of the GNN, it can be seen as embedding a network in the given embedding space; a space where the meaning of each dimension is determined a priori.
> > >
> > > Hopefully this answered your questions. In the chance we might not be understanding your concerns correctly, or you have any other question, do not hesitate to ask, we will always respond to the best of our ability as soon as possible.

---

> > > > ### Comment · Reviewer_LUCZ · 2024-11-27
> > > >
> > > > Thank you! This clears things up. If I understood correctly, this implies that your approach is based on the assumption that the $z$ scores are (ideally) uncorrelated or, at least, only weakly correlated.
> > > >
> > > > However, if one assumes that some $z$ scores are strongly correlated, the hypersphere you describe could collapse along some axes and the non-normalized $z'$ scores might not even lie within a spherical hypervolume. For example, in A.4, you describe that motifs of size 3 are strongly correlated with the 3-paths and triangle occurrence counts are lying on a simplex and not an (Euclidan) sphere. Forcing all signatures to lie on a unit-hypersphere can then result in a distortion of the distribution of occurrence vectors.
> > > >
> > > > From this perspective, my initial question on the validity of L2 normalization, was effectively about whether $z$ scores can be assumed to be (mostly) uncorrelated. I understand that this concern might not be relevant in practice, and that for larger motifs and/or a large number of motifs, any distortions caused by the L2 normalization might be insignificant. If this is the case, this effectively answers my question.
> > > >
> > > > Let me know if I misunderstood your approach and/or its assumptions.
> > > >
> > > > Last, I have a follow-up question/suggestion regarding the evaluation of predicted significance profiles. In your evaluation you used MSE to assess prediction quality. Under the assumption that significance profiles are points on a hypersphere, I believe it would be appropriate to measure the distance between significance profiles via the Cosine distance. Could you explain why you instead chose MSE for the evaluation?
> > > >
> > > > Thank you once again for the insightful discussion!

---

> > > > > ### Author Response · Authors · 2024-11-27
> > > > >
> > > > > Thank you for your interest, time and very interesting remarks!
> > > > >
> > > > > First, just a small note. The strong correlation described in A.4 pertains to Z-scores (normalized or not), not occurrences (number of times a structure appears in a graph). As far as we are concerned, if we do not “anchor” our analysis on control graphs, there is no apparent relation between occurrences.
> > > > >
> > > > > Secondly, you are indeed correct that L2 normalization induces distortion in the distribution of unnormalized Z-scores. The most pronounced example, as you highlighted, occurs in graphs of size 3, where this normalization constrains the values to those discussed in Section 5. However, our approach does not rely on the assumption that Z-scores are uncorrelated. On the contrary, our formulation is grounded in the premise that some correlation exists between scores, as outlined in Section 4.1.
> > > > >
> > > > > Given this, one might reasonably question why we do not predict unnormalized Z-scores directly, considering that normalization can introduce distortion. This concern was addressed in Section 4.1. However, we acknowledge that the potential for distortion was not explicitly discussed in the manuscript. To put it shortly, while correlation between Z-scores does exist, the distortion induced by normalization, as you correctly pointed out, is generally negligible.
> > > > >
> > > > > Furthermore, following seminar work on motif analysis [1] Z-score normalization is essential for enabling fair comparisons across networks of varying sizes. Therefore, the resulting distortion must be accommodated. From our point experience, we maintain that this issue is not significant in most cases; otherwise, we would have to rework the fundamental principles developed under [1] to allow a fairer comparison.
> > > > >
> > > > > In summary, we welcome the presence of some correlation, as it enables our formulation to leverage this relationship. The distortion induced by such correlation is not very severe in most cases, still allowing for a useful motif analysis.
> > > > >
> > > > > ---
> > > > >
> > > > > Regarding your question on the use of MSE over cosine distance: since we operate with normalized scores constrained to the unit hypersphere, MSE becomes proportional to cosine distance. If we were working with unnormalised Z-scores the cosine distance would most likely be the best choice.
> > > > >
> > > > > Again, feel free to add any other question!
> > > > >
> > > > >
> > > > >
> > > > > [1] Ron Milo et al., Superfamilies of Evolved and Designed Networks. Science 303,1538-1542(2004).DOI:10.1126/science.1089167

---

> > > > > > ### Comment · Reviewer_LUCZ · 2024-11-27
> > > > > >
> > > > > > Thank you very much for clarifying! I have no further questions.

---

### Official Review · Reviewer_nUUR · 2024-11-05

**Soundness:** 3
**Presentation:** 2
**Contribution:** 2
**Rating:** 3
**Confidence:** 3

**Summary:**

The authors constructed a large synthetic dataset to test the motif discovery ability of MPNNs, and showed in general that MPNNs do not generalize well beyond synthetic dataset.

**Strengths:**

The authors formulated the motif discovery into a multitarget regression problem on normalized z-scores. And have done an empirical study on a large synthetic dataset to understand the boundaries of MPNNs on motif discovery. The problem they are trying to understand is close to many real world scientific questions (such as motifs in biology).

**Weaknesses:**

I found the conclusions in the paper a bit weaker and satisfying.

The authors spent some time explaining the differences between subgraph counting and motif discovery. The latter can be easier or harder than the subgraph counter depending on the null hypothesis construction. Then the authors tried to answer three key questions (line 402-404), and found the model seems to learn the graph generators and perform better on intra-generator graphs. Following that, the model trained on synthetic data does not generalize well onto real-world graphs. Such findings are pretty expected and I wonder if the readers would learn more from the study.

**Questions:**

The authors mentioned normalizing the z-scores imposes a “mathematical interconnectivity” and further supports multi-target regression tasks, adding an additional layer of interdependence. I wonder if the authors could elaborate on how such a simple constraint has a meaningful impact on the model. Shouldn’t it be a trivial task to learn?

---

> ### Author Response · Authors · 2024-11-24
> **Response to reviewer nUUR**
>
> We thank and appreciate the reviewer for their feedback and response.
>
> 1. A note regarding the weakness pointed out, we may not have communicated our contributions effectively. Our study could be divided into the following parts. We propose ditching the steps from section 3.2. and "(...) design a method for motif finding, leveraging a novel formulation that hinges primarily on reworking the target task to something else other than direct substructure counting." (line 49). We then proceed to elaborate on how this can be achieved (section 4) and analyse if there are any advantages to such reformulation (section 7). We conclude that despite the advantage existing, it is not enough for a 1-WL-limited model trained on synthetic data since it struggles to generalise effectively to more complex, real-world networks. The intent of the analysis of these results was to delve into the underlying reasons behind these outcomes and to offer readers a structured understanding of why such generalisation issues occur and what they represent.
> ---
> 2. If the non-normalized Z-scores were used, any dependencies among the scores become more challenging to detect. Even though it is possible that models could do it, by using normalised Z-scores this task becomes much easier. In fact, by using normalised Z-scores, even humans could discern some dependencies (Figure 3). Hence, since normalising the scores is very easy, using the normalised version as the target was deemed the best option. Moreover, this dependency is so strong that even if a model cannot perfectly infer part $X$ of the graphs in $\Omega$, it can still make reasonable estimates for $X$ based almost entirely on accurate guesses for the other part $Y$. Furthermore, directly obtaining the normalised Z-scores provides stability for cross-network profile comparisons, as the output remains bounded between $-1$ and $1$.

---

### Author Response · Authors · 2024-11-24
**General Response**

We express our sincere appreciation to the reviewers for their invaluable insights.

We also appreciate the reviewers acknowledging that our work “(...) tackles an important and challenging problem (...)” (uysJ), it is “(...) close to many real world scientific questions (...)” (nUUR), and uses a “(...) unique formulation” (DuKG).
We value the recognition by the reviewers of our strategy being “(...) reasonable and seemingly effective.” (QMx2), “(...) well motivated and contextualized (...)” (LUCZ), and the usage of “(...) large synthetic dataset (...)” (nUUR) that “(...) could be useful for subsequent articles on motif discovery.” (QMx2). Finally, we deeply value the recognition that our paper “(...) raises interesting questions (...)” (LUCZ) regarding motif discovery using GNNs.

However, despite some positive comments, after careful consideration, an initial round of discussions, and preliminary evaluations, we have determined that we are unable to submit a revised manuscript that fully addresses the reviewers' many important concerns. Since we cannot adhere to the standards of the conference in the coming timeline, we will formally withdraw the paper in the coming days. Nevertheless, we have provided responses to each review to clarify any potential misunderstandings, answer the questions of the reviewers and to ensure that the paper's review is well-positioned for future endeavours. If the reviewers want to make extra comments, we will promptly respond to them to the best of our ability.

Thank you once again for your time, consideration, and commitment.

---

### Note · Authors · 2024-11-27

**Comment:**

Following our [comment](https://openreview.net/forum?id=PZVVOeu6xx&noteId=NEKUhct1Oe), we confirm the withdrawal of the paper.

**Withdrawal Confirmation:**

I have read and agree with the venue's withdrawal policy on behalf of myself and my co-authors.